SciPost Physics

# Two-dimension to three-dimension transition of chiral spin liquid and fractional quantum Hall phases

Xiaofan Wu and Ya-Hui Zhang$^\star$

William H. Miller III Department of Physics and Astronomy, Johns Hopkins University,
Baltimore, MD 21218, USA
$^\star$ yzhan566@jhu.edu

January 26, 2024

## Abstract

There have been lots of interest in two-dimensional (2D) fractional phases with an emergent $U(1)$ gauge field. However, many experimental realizations are actually in three-dimensional (3D) systems with infinitely stacked 2D layers. Then a natural question arises: starting from the decoupling limit with 2+1d $U(1)$ gauge field in each layer, how does the gauge field become 3+1d when increasing inter-layer coupling? Here we propose a 2D to 3D transition through condensing inter-layer exciton. The Goldstone mode of the condensation becomes the missing $a_z$ component in the 3D phase. We applied this transition mechanism in many different systems.

## 1   Introduction

The study of quantum phase transitions is one of the major focuses in condensed matter physics [1, 2]. Almost all of the well-studied phase transitions are between two phases in the same space-time dimension. In this paper, we are going to consider an unusual class of critical points between a decoupled two-dimensional (2D) phase and a three-dimensional (3D) phase. More specifically, we consider a system with infinitely stacked 2D layers along the $z$ direction, which is quite common in quasi-two-dimensional materials including high-temperature superconducting cuprates and many quantum spin liquid candidates. In this kind of setup, the inter-layer coupling is usually weak, so one can consider the decoupling limit with an independent 2D quantum phase at each layer. If the 2D phase is a fractional phase such as a quantum spin liquid or a fractional quantum Hall (FQH) phase, the inter-layer coupling is usually irrelevant and the decoupled 2D phases survive to a finite inter-layer coupling until a phase transition happens. In the larger coupling regime, the natural ground state should be a three-dimensional phase with excitations mobile in the whole 3D space. The focus of this paper is to describe this kind of 2D to 3D transition.

We will consider the case that the decoupled 2D phase has a $U(1)$ gauge field. Let us take U(1) spin liquid as examples. In the decoupled phase, both the spinon and the emergent photon are confined in each 2D plane. Upon increasing the inter-layer coupling, one can imagine a 3D phase with both spinon and photon moving in the 3D space. Across this 2D to 3D transition, the spinon should get mobile along the $z$-direction, and simultaneously the $U(1)$ gauge field should acquire a missing $a_z$ component with additional Maxwell terms. We will show that both can be accomplished simultaneously through condensing an inter-layer exciton formed by a pair of gauge charges (for example, spinon pairs in spin liquid). Such an exciton condensation $\langle \Phi \rangle \neq 0$ provides a hopping along the $z$-direction for the spinon. Besides, the Goldstone mode of the condensation becomes the missing $a_z$ component of the 3D $U(1)$ gauge field while its phase stiffness provides the missing Maxwell term.

Following this picture, we propose a continuous critical theory for the 2D to 3D transition of a $U(1)$ spin liquid. As a simple illustration, we restrict to the simple chiral spin liquid (CSL) as an example. Chiral spin liquids [3, 4] have been found to be the ground state for various spin 1/2 lattice models [5–17] and also in $SU(N)$ model with $N > 2$ [18–25]. In the simple $SU(2)$ case, it can be thought as a Laughlin state [26] of the spin flips. It is by now well established that the low energy theory describing a CSL or a Laughlin state is through Chern-Simons theory of 2+1d $U(1)$ gauge field [27]. Now we consider a 3D system with infinitely stacked spin layers. When the inter-layer coupling $J_\perp$ is zero, we assume each layer hosts a chiral spin liquid phase. Then we gradually increase $J_\perp$ until a phase transition happens. A natural phase transition is through generating the term $\sum_z \Phi_i(z) f^\dagger_{i;\sigma}(z+1) f_{i;\sigma}(z)$ where $f_{i;\sigma}(z)$ is the spinon in the layer $z$. After the onset of $\Phi$, we have a 3+1d $U(1)$ gauge field, but still with a Chern-Simons term at each layer. This unusual 3D chiral spin liquid turns out to host one gapless mode with quadratic dispersion $\omega \sim q_z^2$ along the $z$ direction and linear dispersion along the $q_x, q_y$ plane.

Next, we study the critical point between the 2D CSL and the gapless 3D CSL. In the small $J_\perp$ side, the $z$ coordinate should have scaling dimension $[z] = 0$ compared to $x, y, t$. In contrast, in the large $J_\perp$ side, we have $[z] = -\frac{1}{2}$ given the $\omega \sim q_z^2$ dispersion. Across the

quantum critical point (QCP), the scaling dimension of the $z$ coordinate needs to jump from 0 to $-1/2$. We will show that it remains zero exactly at the QCP. If we fix $q_x = q_y = 0$, there is gapless mode at every $q_z \in [0, 2\pi]$, coming from the critical boson at each layer. The photon and other order parameters actually acquire a $q_z$ dependence, which leads to a finite but non-zero correlation length along the $z$ direction, indicating a more non-trivial structure than a trivial decoupled fixed point. More specifically, $O^{z_1}(x_1)O^{z_2}(x_2) \sim g_O(z_1 - z_2)\frac{1}{|x_1-x_2|^{\alpha_O}}$, where $x$ denotes the position vector in the $(t, x, y)$ space. For a decoupled fixed point, we expect $g_O(z_1 - z_2) \sim \delta_{z_1, z_2}$. In contrast, our critical theory has $g_O(z_1 - z_2) = e^{-\frac{|z_1-z_2|}{\xi_O}}$, so an operator in one layer correlates with an operator in a layer far away.

Although we focus on the CSL, our theory can be easily generalized to infinitely stacked quantum Hall layers, given the equivalence between the CSL phase and a bosonic Laughlin state. The same construction can lead to a three-dimensional gapless quantum Hall phase. Such a state has been discussed in Ref. [28] from a different construction. Our approach then provides a continuous critical theory between the 3D quantum Hall phases and the decoupled Laughlin states. More recently there have also been discussions of infinite component Chern-Simons-Maxwell (iCSM) theory with both intra-layer and inter-layer chern-simons (CS) terms, with the motivation to construct fracton phases [29–31]. The $U(1)$ gauge field in these phases is still $2 + 1$ d without the $a_z$ component. The inter-layer correlation is encoded through the off-diagonal Chern-Simons term. In contrast, in our construction, the inter-layer correlation is from a Higgs term, which leads to $3 + 1$d $U(1)$ gauge field. It is then natural to explore the case with both inter-layer CS term and inter-layer Higgs condensation. We find that adding a Higgs term from inter-layer exciton condensation to the infinite component Chern-Simons theory always leads to a 3D gapless phase whose low energy spectrum is quite similar to the simple 3D CSL constructed above. Then we can construct a critical theory between a gapped fracton phase [32, 33] described by a iCSM theory and a gapless 3D phase. The critical theory is very similar to the QCP between the 2D CSL and the 3D CSL. We note that criticality out of a fracton phase has also been studied by Ref. [34].

## 2 Transition between 2D and 3D U(1) spin liquid

In this section, we offer a general framework for the 2D to 3D transition of a U(1) spin liquid. We consider the following multi-layer spin model:

$$H = \sum_z \sum_{\langle ij \rangle} J_{ij}\vec{S}_i(z) \cdot \vec{S}_j(z) + ... + \sum_z \sum_i J_\perp \vec{S}_i(z) \cdot \vec{S}_i(z+1) \tag{1}$$

where $z = 0, 1, 2, ..., N-1$ is the coordinate at the $z$ direction, $N$ is the number of layers that will be taken to infinite, $i, j$ are the site indices within each layer, $J_\perp$ is the inter-layer coupling. In ... we include intra-layer terms such as ring exchange terms or chirality terms, which are needed to stabilize a spin liquid phase. As much of this paper is devoted to low-energy field theory, the exact form of the microscopic lattice models is not our focus. Throughout this paper, we assume there is translation invariance along the $z$ direction.

When $J_\perp = 0$, we have decoupled 2D layers. We assume that the ground state is a $U(1)$ spin liquid with emergent $2 + 1$d $U(1)$ gauge field $a_\mu(t, x, y; z), \mu = 0, x, y$ for each layer $z = 0, 1, ..., N-1$. Here the gauge fields in different layers are completely independent. Such a spin liquid phase can be conveniently described by the Abrikosov fermion construction [27]:

$$\vec{S}_i(z) = \frac{1}{2}f^\dagger_{i;\sigma}(z)\vec{\sigma}_{\sigma\sigma'}f_{i;\sigma'}(z) \tag{2}$$

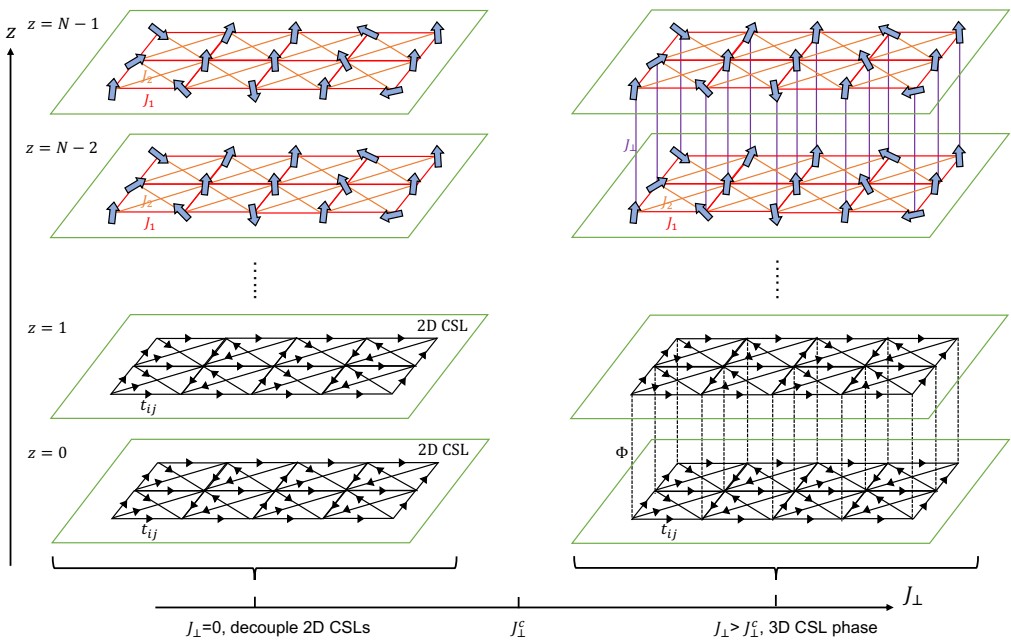

Figure 1: The system we study consists of N identical layers of spin models or quantum Hall layers. We will take $N$ to be infinite in the end. In the example considered in this figure, every layer hosts an independent chiral spin liquid (CSL) at the decoupled limit ($J_\perp = 0$). When $J_\perp > J_\perp^c$, the inter-layer condensate $\Phi \neq 0$ generates a hopping term of the spinons between adjacent layers and leads to a new gapless 3D phase with $3+1$d U(1) gauge field.

with the constraint $\sum_{\sigma=\uparrow,\downarrow} f_{i;\sigma}^\dagger(z) f_{i;\sigma}(z) = 1$ for every $i$ and $z$. There is an emergent U(1) gauge field $a_\mu$ associated with the gauge symmetry: $f_{i;\sigma}(z) \to f_{i;\sigma}(z) e^{i\alpha_i(z)}$.

Let us start from the decoupled phase:

$$\mathcal{L} = \sum_z \mathcal{L}_z[f(z), a(z)] \tag{3}$$

where $z$ is the layer index and $a(z)$ is a $2+1$ d gauge field in layer $z$. $\mathcal{L}_z[f(z), a(z)]$ is the effective action at each layer $z$ which we will specify later. At the decoupling limit $J_\perp \to 0$, $a_\mu(z)$ at different layers fluctuate separately. If we treat the layer index $z$ as the fourth coordinate, we have $a_\mu(x,y,z)$, with $\mu = 0, x, y$. However, there are two essential differences from a true $3+1$ d gauge field: (I) There is no component $a_z(x,y,z)$, which means $b_y$ and $b_x$ cannot be defined [1]. (II) There is only one polarization mode. In the following, we will show that these two problems disappear if we introduce inter-layer exciton condensation.

Suppose there is an onset of an inter-layer exciton condensation $\Phi$ at a critical value of $J_\perp^c$. When $J_\perp > J_\perp^c$, spinons between adjacent layers develop a particle-hole pairing term: $\Phi(z)_i e^{i\theta_i(z)} \sim \langle f_{i;\sigma}^\dagger(z) f_{i;\sigma}(z+1) \rangle \neq 0$. Here $\theta_i(z)$ is the phase of the condensation. The mean-field Hamiltonian now has a new inter-layer hopping term:

$$H_{\text{inter}} = \sum_z \sum_i \Phi_i(z) e^{i\theta_i(z)} f_{i;\sigma}^\dagger(z+1) f_{i;\sigma}(z) + \text{h.c..} \tag{4}$$

Next, we want to learn how the effective low-energy theory changes. In the decoupled theory of each layer, we have gauge transformation: $f_z \to f_z e^{i\chi_z(t,x,y)}$, $a_\mu^z \to a_\mu^z + \partial_\mu \chi_z(t,x,y)$,

---

[1]In continuum theory, $b_y = \partial_z a_x - \partial_x a_z$. If the $a_z$ component is missing, $b_y$ has no gauge independent definition.

where $t$ comes from the path integral construction and the microscopic lattice points are replaced by continuous coordinates $x, y$. Now we have a new condensate field $\Phi_i(z)e^{i\theta_z}$, whose phase should transform as $\theta_z \to \theta_z + \chi_{z+1} - \chi_z$. From this gauge transformation, we can write down the simplest allowed action term for $\theta$ similar to the standard effective theory of a superfluid:

$$S_{\text{int}} = \frac{\rho_s}{2} \int d^3x \sum_z (\partial_\mu \theta_z - (a_\mu^{z+1} - a_\mu^z))^2, \tag{5}$$

and it becomes $f_{\mu 3}f_{\mu 3}$ term in the continuum limit if we see $\theta$ as the fourth component of the vector field $a_z$ ($a_z = \theta_z/b$, $b$ is the inter-layer distance):

$$S_{\text{int}} = \frac{\tilde{\rho}_s}{2b} \int d^4x \sum_{\mu=x,y,z} (\partial_\mu a_z - \partial_z a_\mu)^2. \tag{6}$$

The coupling of $\Phi$ to $f$ is in the form $\Phi_i(z)e^{ia_z}f_i^\dagger(z+1)f_i(z)$, exactly as expected for $a_z$ component of a U(1) gauge field. Thus we obtain a $3+1$ d $U(1)$ gauge theory when $\Phi_z$ condenses.

This transition near $J_\perp = J_\perp^c$ can be described by the onset of the condensation $\Phi$:

$$S = \int d^3x \sum_z \left\{ \sum_{\mu=0,x,y} |(\partial_\mu - i(a_\mu^{z+1} - a_\mu^z))\Phi_z|^2 \right.$$
$$\left. + r|\Phi_z|^2 + \lambda|\Phi_z|^4 + (\Phi_z f_{z+1;\sigma}^* f_{z;\sigma} + \text{h.c.}) \right\} + \sum_z S_f[f_z, a^z], \tag{7}$$

where we assume translation symmetry in the $z$ direction and $\int d^3x$ is integrating over the $t, x, y$ space. $S_f[f_i, a^i]$ is the action of the spinon and gauge field in a single layer, which depends on the ansatz and the type of the spin liquid. The reflection symmetry $R_z : z \to -z$ combined with translation $T_z$ maps $\Phi_i$ to $\Phi_i^\dagger$. So there is an effective particle-hole symmetry for $\Phi$, which forbids the linear $\partial_\tau$ term in the action. When $r < 0$, $\Phi_i$ condenses and the decoupled $2+1$ d $U(1)$ gauge fields develop the fourth component and transits to a $3+1$ d $U(1)$ gauge field as described above.

The above framework works for any U(1) spin liquid no matter whether the spinon is gapped or gapless. In the following sections, we apply it to chiral spin liquid, the simplest 2D spin liquid with a deconfined U(1) gauge field but with gapped matter. In Sec. 3 we constructs a 3+1d CSL following this approach. The critical theory of the 2D to 3D transition of the CSL is discussed in Sec. 4. In Sec. 5 we generalize our theory to an infinite component Chern-Simons theory with inter-layer Chern-Simons terms which may describe a fracton phase. Sec. 6 is the conclusion.

# 3   3+1 d chiral spin liquid

In this section, we study the new $3+1$d CSL phase after the condensation transition. To begin with, here we give a brief introduction to CSL in a single 2D layer. From a spin-$1/2$ model, using the Abrikosov fermion construction in Eq. 2, we can write down a spinon mean-field ansatz for the CSL phase:

$$H_{\text{mean}} = \sum_{\langle ij \rangle} -\frac{1}{2} J_{ij} \left[ (f_{i\sigma}^\dagger f_{j\sigma} t_{ji} + \text{h.c.}) - |t_{ij}|^2 \right] + \sum_i a_0(i)(f_{i\sigma}^\dagger f_{i\sigma} - 1). \tag{8}$$

Here $t_{ij}$ is the spinon pairing which satisfies the self-consistency equation

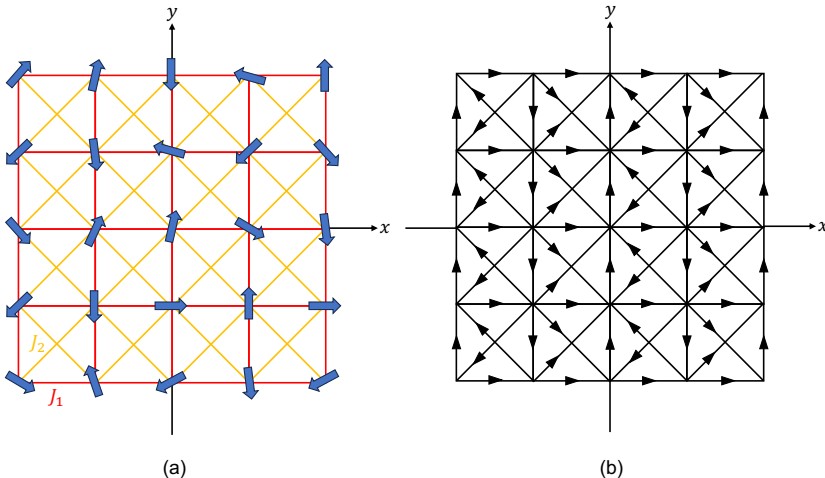

(a)                                                            (b)

Figure 2: (a) The frustrated Heisenberg model of spin-1/2 on the square lattice. $J_1$ and $J_2$ are the nearest and the second nearest coupling constants. (b) The mean-field ansatz of CSL. The hopping $t_{ij}$ has a $\frac{\pi}{2}$ phase in the direction of the arrow. This mean-field ansatz induces $\pi$ flux for each square and $\frac{\pi}{2}$ flux for each triangle.

$$t_{ij} = \left\langle f_{i\sigma}^\dagger f_{j\sigma} \right\rangle, \tag{9}$$

and $a_0(i)$ is determined by the constraint

$$\left\langle f_{i\sigma}^\dagger f_{i\sigma} \right\rangle = 1 \tag{10}$$

at each site. These self-consistent equations may have different solutions of $t_{ij}$ and we call these solutions the mean-field ansatzes. Fig. 2(b) is an example, where we have $t_{i,i+x} = it_1$, $t_{i,i+y} = it_1(-1)^{i_x}$, $t_{i,i+x+y} = t_{i,i+x-y} = -it_2(-1)^{i_x}$, $a_0(i) = 0$. Next, we consider the fluctuations around the mean field. Since the amplitude fluctuation of $t_{ij}$ is gapped, we only consider its phase fluctuation $t_{ij}e^{ia_{ij}}$. We also need to include the fluctuation of $a_0$ since in the path integral formalism it gives the exact constraint Eq. 10. Then we can write down the path integral of the system:

$$Z = \int \mathcal{D}f \, \mathcal{D}[a_0(i)] \mathcal{D}a_{ij} e^{i\int dt L},$$

$$L = \sum_i f_{i\sigma}^\dagger i\partial_t f_{i\sigma} - \left( \sum_{\langle ij \rangle} -\frac{1}{2} J_{ij} \left[ (f_{i\sigma}^\dagger f_{j\sigma} t_{ji} e^{ia_{ji}} + \text{h.c.}) - |t_{ij}|^2 \right] + \sum_i a_0(i)(f_{i\sigma}^\dagger f_{i\sigma} - 1) \right). \tag{11}$$

The fluctuations described by $a_0$ and $a_{ij}$ are actually a U(1) gauge theory. From the Abrikosov fermion construction Eq. 2, we see there is a gauge transformation

$$\begin{aligned}
f_{i\sigma} &\to f_{i\sigma} e^{i\chi_i}, \\
f_{i\sigma}^\dagger &\to f_{i\sigma}^\dagger e^{-i\chi_i}, \\
a_{ij} &\to a_{ij} + \chi_j - \chi_i, \\
a_0 &\to a_0 + \partial_t \chi_i
\end{aligned} \tag{12}$$

that does not change the physical state. In Eq. 11, $a_0$ and $a_{ij}$ behave like external electromagnetic perturbation coupled to the spinon with unit charge. If the ground state is a filled spinon

band with a nonzero Chern number, there should be a Hall conductance. Thus we will get the following Chern-Simons theory in the continuum limit after integrating out spinon $f$:

$$S = \frac{2}{4\pi} \int d^3 x \epsilon^{\mu\nu\rho} a_\mu \partial_\nu a_\rho, \tag{13}$$

where we assume the Chern number of the filled band is 1 and the factor 2 comes from $\sigma = \pm$. This can be realized by the CSL mean-field ansatz in Fig. 2(b). This Chern-Simons theory has an infinitely large gap. To see this, we can add a very small Maxwell term to the theory (since it is irrelevant compared to the Chern-Simons term):

$$S = \frac{2}{4\pi} \int d^3 x \epsilon^{\mu\nu\rho} a_\mu \partial_\nu a_\rho - \frac{1}{4g^2} \int d^3 x f_{\mu\nu} f^{\mu\nu}. \tag{14}$$

Note that Eq. 14 is written in Minkovski space-time. In most sections throughout this paper (except for Sec. 3.1) we are using the Euclidean space-time action. So we also write down the Euclidean space-time counterpart of Eq. 14 here:

$$S = \frac{2i}{4\pi} \int d^3 x \epsilon_{\mu\nu\rho} a_\mu \partial_\nu a_\rho + \frac{1}{4g^2} \int d^3 x \left( \partial_\mu a_\nu - \partial_\nu a_\mu \right)^2, \tag{15}$$

where $\mu, \nu, \rho$ are summed over $0, 1, 2$, $\epsilon^{012} = -\epsilon_{012} = 1$.

By solving the equation of motion, we get a single photon mode with an energy gap

$$E_{\text{gap}} = \frac{g^2}{\pi}, \tag{16}$$

which goes to infinity in the limit $g^2 \to \infty$.

Next, we turn on the interlayer coupling $J_\perp$ and let the system go through the transition. Now we have a new phase variable $\theta$ of the condensate $\Phi$ to be the $a_z$ component. As we shall see, a new gapless photon mode shows up in the new 3+1 d CSL.

## 3.1 Gapless photon modes in the 3D CSL

After the condensation transition, in addition to the mean-field Hamiltonian Eq. 8 for each layer, there is a new spinon hopping term in the Hamiltonian which allows the spinon to hop between different layers. The mean field Hamiltonian now is:

$$
\begin{aligned}
H &= H_{\text{mean}} + H_{\text{inter}} \\
&= \sum_z \sum_{\langle ij \rangle} -\frac{1}{2} J_{ij} \left[ (f_{i;\sigma}^\dagger(z) f_{j;\sigma}(z) t_{ji} e^{i a_{ji}(z)} + \text{h.c.}) - |t_{ij}|^2 \right] \\
&\quad + \sum_i a_0^z(i)(f_{i;\sigma}^\dagger(z) f_{i;\sigma}(z) - 1) + \sum_z \sum_i \Phi_i(z) e^{i a_z^z} f_{i;\sigma}^\dagger(z+1) f_{i;\sigma}(z) + \text{h.c.}.
\end{aligned}
\tag{17}
$$

where $a_z^z$ is the phase of the condensation $\Phi_z(z)$.

Integrating out the spinon, we get the low energy effective theory of phase fluctuations:

$$
\begin{aligned}
S &= \frac{2i}{4\pi} \sum_z \int d^3 x \epsilon_{\mu\nu\rho} a_\mu^z \partial_\nu a_\rho^z + \frac{1}{4g^2} \sum_z \int d^3 x (\partial_\mu a_\nu^z - \partial_\nu a_\mu^z)^2 \\
&\quad + \frac{\rho_s}{2} \sum_z \int d^3 x \left( \partial_\mu a_z^z - \left( a_\mu^{z+1} - a_\mu^z \right) \right)^2.
\end{aligned}
\tag{18}
$$

164    We can take the continuum limit in the $z$-direction in the equation above and get a contin-
165 uum model of the 3+1 d CSL. Note that under the continuum limit, the second line becomes
166 the missing Maxwell term $(\partial_\mu a_z - \partial_z a_\mu)^2$. The Minkovski action in the continuum limit is as
167 follows:

$$S = \frac{1}{b}\int \mathrm{d}^4 x \left(-\frac{1}{4g^2}f_{\mu\nu}f^{\mu\nu} - \frac{\tilde{\rho}_s}{2}f_{\mu 3}f^{\mu 3} + \frac{k}{4\pi}\epsilon^{\mu\nu\rho}a_\mu\partial_\nu a_\rho\right), \tag{19}$$

168 where $b$ is the inter-layer distance, $k = 2$ is the integer level in Chern-Simons theory, $\tilde{\rho}_{\mathbf{s}} = \rho_s b^2$,
169 $\mu, \nu, \rho$ run over 0, 1, 2. The first term is the $2 + 1$ d Maxwell term. The coefficient of $f_{\mu 3}f^{\mu 3}$
170 is different from the first term since it is generated by the condensation mechanism. The third
171 term is the Chern-Simons term.

172    We can use the variational principle $\delta S = 0$ to get the classical equation of motion. In
173 Maxwell's theory, this gives us the inhomogeneous part of the Maxwell's equations (we do not
174 include sources in the action for the moment). The homogeneous part does not change. So
175 we have a new set of "Maxwell's equations":

$$\nabla \cdot \vec{e} + (\tilde{\rho}_s g^2 - 1)\partial_z e_z - \frac{kg^2}{2\pi}b_z = 0, \tag{20}$$

$$\partial_t \vec{e} - \nabla \times \vec{b} - (\tilde{\rho}_s g^2 - 1)\partial_z(\hat{z}\times\vec{b}) - \frac{kg^2}{2\pi}\hat{z}\times\vec{e} = 0, \tag{21}$$

$$\nabla \cdot \vec{b} = 0, \tag{22}$$

$$\nabla \times \vec{e} + \partial_t \vec{b} = 0. \tag{23}$$

176    Now we can find the plane wave solutions $\vec{e} = \mathcal{E}e^{i\mathbf{q}\cdot\mathbf{x}-i\omega t}$, $\vec{b} = \mathcal{B}e^{i\mathbf{q}\cdot\mathbf{x}-i\omega t}$ to the equations
177 above. Details are in Appendix A). We get two photon modes $\mathcal{B}_\pm$ (we choose to use the
178 magnetic field) with dispersion relations

$$\omega_\pm^2 = q_x^2 + q_y^2 + \tilde{\rho}_{\mathbf{s}}g^2 q_z^2 + \frac{k^2 g^4}{8\pi^2} \pm \frac{k^2 g^4}{8\pi^2}\sqrt{1 + \frac{16\pi^2\tilde{\rho}_{\mathbf{s}}}{k^2 g^2}q_z^2} \tag{24}$$

179    We can see that $\mathcal{B}_+$ has an energy gap $\frac{kg^2}{2\pi}$ which goes to infinity as $g^2 \to \infty$, while $\mathcal{B}_-$
180 is a gapless mode with $\omega_-^2 = q_x^2 + q_y^2 + \frac{4\pi^2\tilde{\rho}_{\mathbf{s}}^2}{k^2}q_z^4 + \cdots$ when $q_z$ is small. $\mathcal{B}_\pm$ are elliptically
181 polarized with opposite circular direction. In other words, the Chern-Simons term will pick up
182 a preferred circular direction. When the wave vector $\mathbf{q}$ lies in $x-y$ plane, we can see that $\mathcal{B}_\pm$
183 both become linearly polarized: the gapped mode $\mathcal{B}_+$ is in $z$ direction while the new gapless
184 mode $\mathcal{B}_-$ lies in $x-y$ plane. Remember that in 2+1 d Chern-Simons theory, there is only one
185 gapped mode and the magnetic field only has a z-component. Here $\mathcal{B}_+$ looks very similar to
186 that mode: it is linearly polarized in the z-direction and has a large energy gap which goes
187 to infinity as $g^2 \to \infty$ as in $2 + 1$ d Chern-Simons theory. The other gapless mode $\mathcal{B}_-$ lies
188 in the x-y plane, which cannot exist unless we have the fourth component of the gauge field.
189 So starting from the 2D CSL, the gapped photon mode remains gapped across the transition,
190 while a new gapless mode emerges only after the transition.

## 3.2 Alternative derivation from Higgs mechanism and its equivalence to $a_z = 0$ gauge

193 When the number of layers $N$ is finite, one can treat our theory as purely $2 + 1$d. In $2 + 1$d
194 theory, the phase $\theta$ of the condensate $\Phi$ is just like the Goldstone mode in Higgs-mechanism.
195 So how do we understand the gapless photon mode in the Higgs language?

196    Instead of treating $\theta$ as $a_z$ to get the continuum $3+1$ d $U(1)$ gauge theory, we can also
197  integrate it out to get a $2+1$ d $U(1)$ gauge theory. This approach is essentially the same as
198  using the $a_z = 0$ gauge of the $3+1$ d gauge field. Integrating out $\theta$ in Eq.(5) gives a additional
199  term $S_{\text{int}}$:

$$
\begin{aligned}
S_{\text{int}} &= \frac{\rho_s}{2} \sum_z \int d^3x \left(a_\mu^{z+1,\perp} - a_\mu^{z,\perp}\right)^2 \\
&= \frac{1}{2} \sum_{q_z} \int \frac{d^3q}{(2\pi)^3} u(q_z) a_\mu^{q_z}(q) \left(\delta_{\mu\nu} - \frac{q_\mu q_\nu}{q^2}\right) a_\nu^{-q_z}(-q),
\end{aligned}
\tag{25}
$$

200  where $a_\mu^\perp$ is the transverse component of the gauge field satisfying $\partial_\mu a_\mu^\perp = 0$ [35], $\mu, \nu = 0, 1, 2$,
201  $q_z = 0, \frac{2\pi}{N}, \ldots \frac{2\pi(N-1)}{N}$ is the discrete momentum in $z$-direction, the coefficient

$$
u(q_z) = 4\rho_s \sin^2(q_z/2) \tag{26}
$$

202  is $q_z$ dependent. The subscript "int" stands for inter-layer condensation. The above action is
203  just the familiar Higgs mass for U(1) gauge field which however is $q_z$ dependent.
204    Then the $3+1$ d CSL is described by the action

$$
\begin{aligned}
S &= \frac{ik}{4\pi} \sum_z \int d^3x \epsilon_{\mu\nu\rho} a_\mu^z \partial_\nu a_\rho^z + \frac{1}{4g^2} \sum_z \int d^3x (\partial_\mu a_\nu^z - \partial_\nu a_\mu^z)^2 + S_{\text{int}} \\
&= \frac{1}{2} \sum_{q_z} \int \frac{d^3q}{(2\pi)^3} a_\mu^{q_z}(q) \left[\frac{k}{2\pi} \epsilon_{\mu\rho\nu} q_\rho + \left(\frac{q^2}{g^2} + u(q_z)\right) \left(\delta_{\mu\nu} - \frac{q_\mu q_\nu}{q^2}\right)\right] a_\nu^{-q_z}(-q) \\
&= \frac{1}{2} \sum_{q_z} \int \frac{d^3q}{(2\pi)^3} a_\mu^{q_z}(q) (D^{-1})_{\mu\nu}^{q_z}(q) a_\nu^{-q_z}(-q).
\end{aligned}
\tag{27}
$$

205  Here we work in imaginary time so the Chern-Simons term is imaginary. $q = (q_0, q_x, q_y)$ is the
206  $2+1$ d wave vector and $q^2 = q_0^2 + q_x^2 + q_y^2$. $g^2$ is a large coupling constant. We can inverse the
207  matrix in the transverse subspace to get the photon propagator:

$$
D_{\mu\nu}^{q_z}(q) = \frac{-2\pi/k}{q^2 + \frac{4\pi^2}{k^2}(u(q_z) + \frac{q^2}{g^2})^2} \epsilon_{\mu\nu\rho} q_\rho + \frac{\frac{4\pi^2}{k^2}(u(q_z) + \frac{q^2}{g^2})}{q^2 + \frac{4\pi^2}{k^2}(u(q_z) + \frac{q^2}{g^2})^2} \left(\delta_{\mu\nu} - \frac{q_\mu q_\nu}{q^2}\right). \tag{28}
$$

208  The dispersion relations are given by its poles,

$$
\omega_\pm^2 = \mathbf{q}^2 + g^2 u(q_z) + \frac{k^2 g^4}{8\pi^2} \pm \frac{k^2 g^4}{8\pi^2} \sqrt{1 + \frac{16\pi^2 u(q_z)}{k^2 g^2}}, \tag{29}
$$

209  where $\mathbf{q}^2 = q_x^2 + q_y^2$. This is the discrete version of Eq.(24). We can see that the $\omega_+$ mode has
210  an energy gap $\frac{kg^2}{2\pi}$ which goes to infinity as $g^2 \to \infty$, while at small $q_z$, $\omega_-^2 = \mathbf{q}^2 + \frac{4\pi^2 \rho_s^2 q_z^4}{k^2}$ is
211  gapless.
212    If $N$ is finite, then $q_z = \frac{2\pi}{N} j, j = 0, 1, .., N-1$ is discrete. Then we find that only the
213  $q_z = 0$ mode is gapless while the other modes are all gapped. This is in agreement with our
214  expectations. Considering a 2D system with $N$ number of layers, condensation of $\Phi$ just locks
215  the U(1) gauge fields from different layers together, while other components acquire a mass
216  term. However, when $N$ approaches infinite, the gap of other components decreases as $\frac{1}{N^2}$ and
217  we need to view the system as 3D above this small energy scale.

## 4 Critical properties of the 2D to 3D transition

In the last section, we discussed the properties of the 3D CSL phase after the condensation transition. In this section, we are going to discuss the critical point at $J_\perp = J_\perp^c$. In Eq. 7, when the spinon fulfills the ansatz for CSL (see Fig. 2), we can integrate out the spinon field and get the following action:

$$
S = \sum_z \int d^3x |(\partial_\mu - i(a_\mu^{z+1} - a_\mu^z))\Phi_z|^2 + \frac{i\alpha}{4\pi} \sum_z \int d^3x \epsilon_{\mu\nu\rho} a_\mu^z \partial_\nu a_\rho^z
$$
$$
+ s|\Phi|^2 + \frac{1}{2} \sum_{z,z'} \lambda_{z,z'} \int d^3x |\Phi_z|^2 |\Phi_{z'}|^2.
\tag{30}
$$

Here $\Phi_z$ is a complex boson between layer $z$ and $z+1$, $\alpha = 2$ since we have spin $\sigma = \uparrow, \downarrow$. In the $|\Phi|^4$ term we assume $\lambda_{z,z'}$ has translational symmetry along the $z$ direction. Note that we have a critical boson $\Phi_z$ at each layer $z$ and the gauge field in the action is $2+1$d. We list the gauge invariant physical operators at both the critical point and the 3D CSL phase in Table.(1). We can see that the new field strength $b_3, e_1, e_2$ in the 3D phase developed from the current operator related to the phase of $\Phi$ at the critical point. When $s > 0$, this mode is gapped. When $s < 0$, the phase of $\Phi$ becomes the gapless photon mode in the 3D phase.

We note that the gauge symmetry forbids $\Phi_z^\dagger \Phi_{z+1}$ term, so the critical bosons $\Phi_z$ from different layers do not hybridize. However, the Higgs boson at one layer can interact with the boson at another layer through the photon $a_\mu$. Although the U(1) gauge field $a_\mu$ is still $2+1$d in the sense that there is only $\mu = 0, x, y$ component, we will see that the photon acquires a $q_z$ dependence. But the $q_z$ dependence is not through the usual dispersion: the photon energy is zero for any $q_z$ as long as $\mathbf{q} = 0$. In the following, we use $\mathbf{q}$ to indicate the momentum in the $x, y$ plane. In the end, our critical theory has infinite gapless critical modes coming from the layer structure, but it is not in a trivial layer decoupled fixed point. More specifically, the correlation function has the form $O^{z_1}(x_1)O^{z_2}(x_2) \sim g_O(z_1 - z_2)\frac{1}{|x_1 - x_2|^{\alpha_O}}$, where $x$ denotes the coordinate vector in the $(t, x, y)$ plane. For a decoupled fixed point, we expect $g_O(z_1 - z_2) \sim \delta_{z_1, z_2}$. In contrast, our critical theory has $g_O(z_1 - z_2) = e^{-\frac{|z_1 - z_2|}{\xi_O}}$. When $s < 0$, the $z$ coordinate becomes normal and we can take the continuum limit. However, at $s = 0$ we need to maintain the layer structure in the theory and keep the modes from each $q_z \in [0, 2\pi)$.

In order to do controlled perturbative calculation, we use the large $N_b$ expansion [36] to study the critical behavior at the transition point. We assume there are $N_b$ flavors of bosonic fields $\Phi_z^a$ at each layer, which also means that we have $N_b$ flavors of spinon since $\Phi$ represents spinon pairing so we should make substitution $\alpha \to N_b \alpha$ in the Chern-Simons term. The action now becomes:

$$
S = \sum_z \sum_{a=1}^{N_b} \int d^3x |(\partial_\mu - i(a_\mu^{z+1} - a_\mu^z))\Phi_z^a|^2 + \frac{iN_b\alpha}{4\pi} \sum_z \int d^3x \epsilon_{\mu\nu\rho} a_\mu^z \partial_\nu a_\rho^z
$$
$$
+ \frac{1}{2} \sum_{z,z'} \sum_{a,b=1}^{N_b} \int d^3x \lambda_{z,z'} |\Phi_z^a|^2 |\Phi_{z'}^b|^2.
\tag{31}
$$

In many circumstances, the Chern-Simons term has been shown to have no RG flowing up to two-loop level [37, 38]. So it is a good guess to assume $\alpha = 2$ in this action. The mass term for $\Phi$ is tuned to be zero. In addition, in writing down the quartic $\Phi^4$ term, we have assumed the $SU(N_b)$ symmetry at each layer is preserved at the critical point. $\lambda_{z,z'}$, which is empirically of order $1/N_b$, may have a specific form but should have translational invariance. Its exact

| Operators at the critical point | | Operators in the 3D phase |
|---|---|---|
| $\|\Phi_z\|^2 \sim \vec{S}_i(z) \cdot \vec{S}_i(z+1)$ | | |
| $\partial_\mu a_\nu - \partial_\nu a_\mu \sim b_3, e_1, e_2$ | $\Longleftrightarrow$ | $\partial_\mu a_\nu - \partial_\nu a_\mu \sim b_3, e_1, e_2$ |
| $2\mathrm{Im}\left\{\Phi_z^* \left(\partial_\mu - i\left(a_\mu^{z+1} - a_\mu^z\right)\right)\Phi_z\right\}$ $\sim$ Current $J_\mu$ | $\Longleftrightarrow$ | $\partial_\mu a_z - \partial_z a_\mu \sim b_1, b_2, e_3$ |

Table 1: Physical quantities and their associated operators at both the critical point and the 3D phase.

form, as we will see, is not important as long as the first order correction is concerned. The bare photon propagator is (throughout this paper, we use the Landau gauge[2])

$$D_{0,\mu\nu}^{z,z'}(q) = -\frac{2\pi}{N_b \alpha} \frac{\epsilon_{\mu\nu\lambda} q_\lambda}{q^2} \delta^{z,z'} = D_{0,\mu\nu}(q)\delta^{z,z'}. \tag{32}$$

We introduce a new field variable $\alpha_\mu^z = a_\mu^{z+1} - a_\mu^z$ and use it in the Feynman diagram calculation. We may also call it photon in the following. Its bare propagator is

$$\tilde{D}_{0,\mu\nu}^{z,z'}(q) = D_{0,\mu\nu}(q)(2\delta^{z,z'} - \delta^{z,z'+1} - \delta^{z,z'-1}). \tag{33}$$

Finally, we do a Hubbard-Stratonovich (HS) transformation and introduce a bosonic field $\varphi_z$ to decompose the $\Phi^4$ term. The action is given below, where we leave the quadratic parts of $\varphi$ and $\alpha$ since we are to use their large $N_b$ effective propagators in the Feynman diagram calculation.

$$S = \sum_z \sum_{a=1}^{N_b} \int \mathrm{d}^3x |(\partial_\mu - i\alpha_\mu^z)\Phi_z^a|^2 + \sum_z \sum_{a=1}^{N_b} \int \mathrm{d}^3x \varphi_z |\Phi_z^a|^2. \tag{34}$$

Figure 3: Bubble diagrams for the effective propagators of the gauge field and the HS field $\varphi$. The bare propagators are of order $1/N_b$ and each boson loop (blue circle) has a factor of $N_b$, so all the bubble diagrams are of the same order $1/N_b$. (a) Black wavy lines represent the bare photon propagator $\tilde{D}_0$ (Eq. 32 and Eq. 33). The red wavy line represents the effective photon propagator $\tilde{D}_{\mathrm{eff}}$. (b) Black dashed lines represent the bare propagator $G_{\varphi,0}^{z,z'} = -\lambda_{z,z'}$. The red dashed line represents the effective propagator $G_{\varphi,\mathrm{eff}}$.

In Appendix. B we present the calculation of the effective propagators of the gauge field and the scalar $\varphi$. Due to the translational invariance in $z$-direction, we can Fourier transform

[2]This can be done by adding a gauge fixing term $(1/2\xi)(\partial_\mu a_\mu^z)^2$ to get rid of the zero eigen-value problem when doing the matrix inverse, and then take the limit $\xi \to 0$ in the resulting propagator.

the layer index into $q_z$. In the following we use $q_\mu, p_\mu$ for the momentum in the $(0, x, y)$ subspace. $q_z$ is used as an additional index. The propagators for the gauge field $\alpha$ and the scalar $\varphi$ are

$$\tilde{D}^{q_z}_{\text{eff},\mu\nu}(q) = \frac{A(q_z)}{N_b}\left(\frac{B(q_z)}{|q|}\left(\delta_{\mu\nu} - \frac{q_\mu q_\nu}{q^2}\right) - \frac{\epsilon_{\mu\nu\lambda}q_\lambda}{q^2}\right) + \mathcal{O}(1/N_b^2), \tag{35}$$

$$G^{q_z}_{\varphi,\text{eff}}(p) = -\frac{8|p|}{N_b} + \mathcal{O}(1/N_b^2), \tag{36}$$

where we introduced two $q_z$ dependent functions $A(q_z)$ and $B(q_z)$ as follows:

$$A(q_z) = \frac{\frac{8\pi}{\alpha}\sin^2\frac{q_z}{2}}{1 + \frac{\pi^2 \sin^4 \frac{q_z}{2}}{4\alpha^2}}, \quad B(q_z) = \frac{\pi \sin^2 \frac{q_z}{2}}{2\alpha}. \tag{37}$$

We can also get the propagator of the original gauge field $a_\mu$ using:

$$D^{q_z}_{\text{eff},\mu\nu}(q) = \frac{\tilde{D}^{q_z}_{\text{eff},\mu\nu}(q)}{4\sin^2\frac{q_z}{2}}, \tag{38}$$

which also shows $q_z$ dependence. This is because the condensate field $\Phi_z$ couples to $a_\mu^{z+1} - a_\mu^z$ and therefore the bubble diagrams Fig. 3 connect $a_\mu$ at different layers.

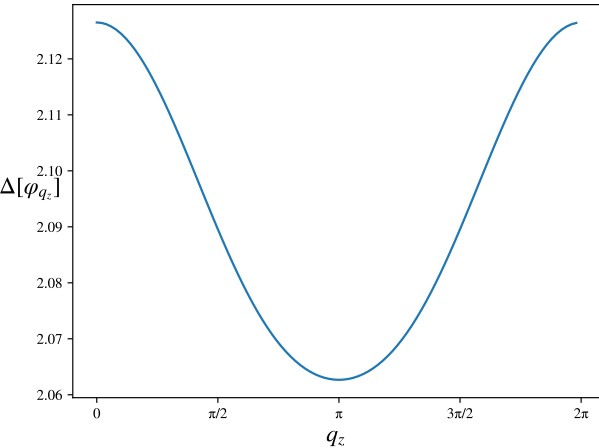

Figure 4: Scaling dimension of $\varphi_{q_z}$ given by Eq. 40 and Eq. 41. We choose $N_b = 2$, $\alpha = 2$, and take $N \to \infty$ so the momentum summation is replaced by an integral.

On the contrary, there is no $q_z$ dependence in the leading order for $\varphi$. This is because $\varphi_z$ couples to $|\Phi_z^a|^2$ and the bubble diagrams in Fig. 3 only connect $\varphi_z$ at the same layer. So the effective propagator $G^{q_z}_{\varphi,\text{eff}}$ is $q_z$ independent. However, $G_\varphi$ will still acquire a $q_z$ dependence in the next order of $1/N_b$.

We calculate the scaling dimension of $\varphi$ up to order $1/N_b$ using the same techniques in Ref. [36]. Basically, we calculate the logarithmic divergent part of the 2-point correlation function $\langle \varphi_z(x)\varphi_{z'}(0)\rangle$ and then reexponentiate it. The results are summarized in Table. 2 and the detailed calculation is in Appendix. C. It turns out that the scaling dimensions of $\varphi_z$ at each layer are mixed and we need to go to the $q_z$ space and find out the independent scaling dimensions of the operators $\varphi_{q_z}(x) = \frac{1}{\sqrt{N}}\sum_z e^{-iq_z \cdot z}\varphi_z(x)$. Then we sum over the results in Table. 2, Fourier transform it into $q_z$ space, and reexponentiate it. Finally we obtain the 2-point correlation function

A.   $x, z$ $\text{----}$ $0, z'$   $= \frac{8}{\pi^2 N_b |x|^4} \delta_{z,z'} = U \delta_{z,z'}$

B.   $= -2 \times \frac{2}{3\pi^2 N_b} \cdot \frac{1}{N} \sum_{q_z} A(q_z) B(q_z) \ln(x^2 \Lambda^2) U \delta_{z,z'}$

C.   $= 0$

D.   $= 2 \times \frac{2}{3\pi^2 N_b} \ln(x^2 \Lambda^2) U \delta_{z,z'}$

E.   $= \frac{4}{\pi^2 N_b} \ln(x^2 \Lambda^2) U \delta_{z,z'}$

F.   $= \frac{1}{4\pi^2 N_b} \cdot \frac{1}{N} \sum_{q_z, l_z} e^{i q_z \cdot (z - z')} A(l_z) A(q_z - l_z)$
$\cdot (B(l_z) B(q_z - l_z) - 1) \ln(x^2 \Lambda^2) U$

G.   $= 0$

H.   $= 0$

I.   $= 0$

Table 2: Results for the individual Feynman diagrams appearing in the first order correction to the 2-point correlation function $\langle \varphi_z(x) \varphi_{z'}(0) \rangle$. We only calculate the logarithmic divergent part of each diagram and zero means no logarithmic divergence.

$$G_\varphi^{q_z}(x) = \langle \varphi_{q_z}(x)\varphi_{-q_z}(0)\rangle = \left(\frac{8}{\pi^2 N_b |x|^4}\right)\left(\frac{1}{x^2\Lambda^2}\right)^{\Delta_{q_z}^{(1)}}, \tag{39}$$

283 where $x = (t, x, y)$, $x^2 = t^2 + x^2 + y^2$, $\Lambda$ is a momentum cutoff, $\Delta_{q_z}^{(1)}$ is the anomalous
284 dimension of $\varphi_{q_z}$ at order $1/N_b$,

$$\Delta_{q_z}^{(1)} = \frac{4}{3\pi^2 N_b}\frac{1}{N}\sum_{l_z} A(l_z)B(l_z) - \frac{16}{3\pi^2 N_b}$$
$$- \frac{1}{4\pi^2 N_b}\frac{1}{N}\sum_{l_z} A(l_z)A(q_z - l_z)(B(l_z)B(q_z - l_z) - 1). \tag{40}$$

285 The scaling dimension of $\varphi$ is

$$\Delta[\varphi_{q_z}] = 2 + \Delta_{q_z}^{(1)} + \mathcal{O}(1/N_b^2). \tag{41}$$

286 We also show the numerical result of $\Delta[\varphi_{q_z}]$ for infinite-layer case ($N \to \infty$) in Fig. 4.

## 4.1   correlation in $z$-direction

288 Note that $\varphi$ is a gauge invariant operator, it corresponds to a physical observable

$$|\Phi(z)|^2 \sim \vec{S}_i(z) \cdot \vec{S}_i(z+1). \tag{42}$$

289 We can compute its spectral weight

$$S_\varphi(\omega, q_z, \mathbf{q}) = -2\mathrm{Im}G_\varphi^{q_z}(q)|_{i\omega \to \omega + i0+}, \tag{43}$$

290 where $G_\varphi^{q_z}(q)$ is the Fourier transform of Eq.(39). The result is

$$S_\varphi(\omega, q_z, \mathbf{q}) \sim \frac{1}{N_b}\Theta(|\omega| - |\mathbf{q}|)\,\mathrm{sign}(\omega)(\omega^2 - |\mathbf{q}|^2)^{\frac{1}{2}+\Delta_{q_z}^{(1)}}\cos\left(\pi\Delta_{q_z}^{(1)}\right). \tag{44}$$

291 It shows 'local criticality' along $z$-direction, in the sense that $S(\omega, q_z, \mathbf{q} = 0)$ has zero energy
292 excitation in the whole range of $q_z \in [0, 2\pi]$. This means that we cannot do scaling and RG
293 flow of $q_z$ direction at all.

294     By Fourier transforming Eq. 39, we can learn about how the correlation function of $\varphi_z$
295 decays in $z$-direction. Notice that Eq. 39 is only valid in a large distance of $x$, and we have
296 little knowledge about the UV physics at $x = 0$ limit. Our strategy is to fix a large but finite
297 $x$, and then see how the correlation function $G_\varphi^{z-z'}(x)$ vary as we increase $z - z'$. Notice that
298 in Eq. 39, the $q_z$ dependence is reflected in the power of $1/x^2\Lambda^2$. So we cannot obtain the
299 asymptotic form $G_\varphi^{z-z'}(x) \sim g_\varphi(z-z')\frac{1}{|x|^{\alpha_\varphi}}$, since the dependence on $z$ and $x$ are not separated.
300 What we can do is the following integral

$$g\left(z - z'; x\Lambda\right) = \frac{1}{2\pi}\int_0^{2\pi} dq_z e^{iq_z \cdot (z-z')}\left(\frac{1}{x^2\Lambda^2}\right)^{\Delta_{q_z}^{(1)}} \tag{45}$$

301 at a fixed $1/x^2\Lambda^2$. The numerical result is in Fig. 5. We can see that at a fixed $x$, the correlation
302 function $G_\varphi^{z-z'}$ exponentially decay in $z$ direction. However, the correlation length also depends
303 on $x$. $G_\varphi^{z-z'}(x)$ now is

$$G_\varphi^{z-z'}(x) = g\left(z - z'; x\Lambda\right)\left(\frac{8}{\pi^2 N_b |x|^4}\right). \tag{46}$$

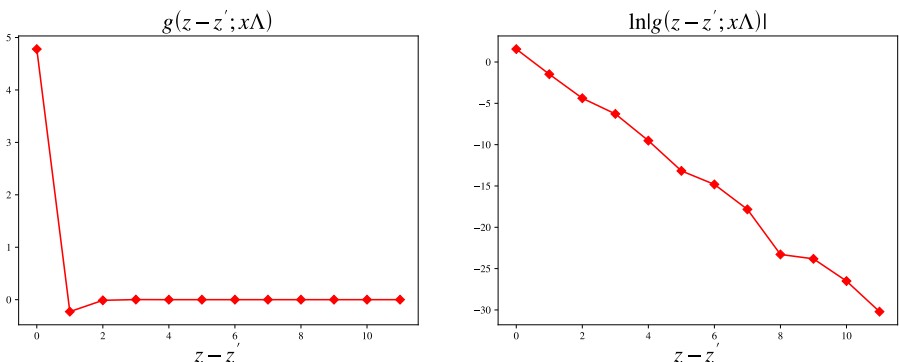

Figure 5: The numerical result of $g\left(z - z'; x\Lambda\right)$ in Eq. 45, which characterizes the decay of the correlation function of $|\Phi|^2$ in $z$-direction for a fixed $x$. We have chosen the parameters $N_b = 2$, $\alpha = 2$ and $(1/x^2\Lambda^2) = 0.05$. The logarithm of $g\left(z - z'; x\Lambda\right)$ is also plotted which shows the exponential decay.

Correlation functions of other physical operators can also be studied. We list the physical quantities and their expressions at both the critical point and the 3D phase in Table. 1. For example, consider the correlation function of $b_3 = \partial_1 a_2 - \partial_2 a_1$. Using the effective propagator Eq. 38, we can see that the leading order is already $q_z$ dependent and is as follows:

$$\left\langle b_3^{q_z}(q) b_3^{-q_z}(q) \right\rangle = \left( \frac{1}{N_b} \cdot \frac{\pi^2 \sin^2(\frac{q_z}{2})}{\alpha^2 + \frac{1}{4}\pi^2 \sin^4(\frac{q_z}{2})} \right) \frac{q_1^2 + q_2^2}{|q|}. \tag{47}$$

For the correlation function of $b_3$, the $q_z$ dependence is in a prefactor separated from the $q$ dependence. So when we Fourier transform it to real space, the Fourier transformations into $z$ and into $x$ are independent of each other:

$$\left\langle b_3^z(x) b_3^{z'}(0) \right\rangle = g_{b_3}(z - z') \mathcal{F}_{b_3}(x), \tag{48}$$

$$g_{b_3}(z - z') = \frac{1}{2\pi} \int_0^{2\pi} dq_z e^{iq_z \cdot (z - z')} \frac{1}{N_b} \cdot \frac{\pi^2 \sin^2(\frac{q_z}{2})}{\alpha^2 + \frac{1}{4}\pi^2 \sin^4(\frac{q_z}{2})}, \tag{49}$$

where $\mathcal{F}_{b_3}(x) \sim 1/x^4$ by dimensional analysis. The numerical result of $g_{b_3}(z - z')$ is in Fig. 6. We can see that the correlation function $b_3$ exponentially decays in $z$-direction. This time, the correlation length in $z$-direction is independent of $x$.

# 5 Continuous transition between gapped fracton order and gapless 3D phase

Our 2D to 3D transition of CSL can be easily generalized to the fractional quantum Hall phase. For example, the same theory (with a different level of the Chern-Simons term) can describe a transition between decoupled 1/3 Laughlin state and a gapless 3D quantum Hall phase proposed in Ref. [28]. In this section, we try to make a more non-trivial generalization.

## 5.1 2D iCSM theory

The decoupled CSL or Laughlin state is described by a K matrix with dimension $N \times N$, where $N$ as usual is the number of layers. For the decoupled CSL or Laughlin state, the K matrix only

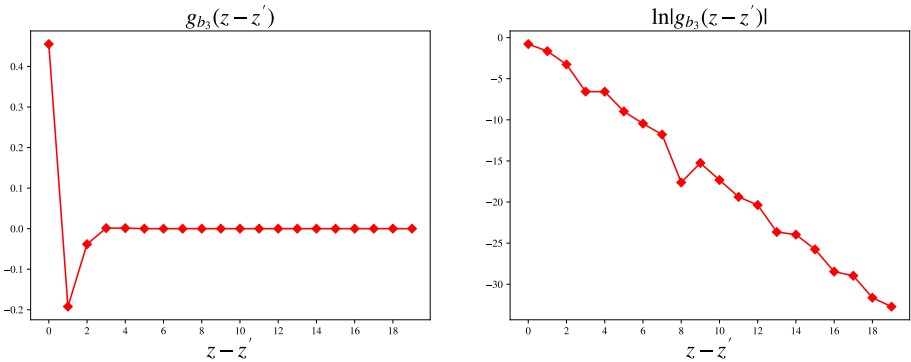

Figure 6: The numerical result of $g_{b_3}\left(z-z'\right)$ in Eq. 49, which characterizes the decay of the correlation function of $b_3$ in $z$-direction. We have chosen the parameters $N_b = 2$, $\alpha = 2$. The logarithm of $g_{b_3}\left(z-z'\right)$ is also plotted which shows the exponential decay.

has diagonal elements. But one can easily imagine a phase with also off-diagonal elements in the $N \times N$ K matrix. A more general $K$ matrix was discussed previously [29–31] and called infinite component Chern-Simons-Maxwell(iCSM) theory. The effective action is

$$S_{\text{2D,iCSM}} = \frac{i}{4\pi} \sum_{z,z'} K_{z,z'} \int \mathrm{d}^3 x \, \epsilon_{\mu\nu\rho} a_\mu^z \partial_\nu a_\rho^{z'} + \frac{1}{4g^2} \sum_z \int \mathrm{d}^3 x (\partial_\mu a_\nu^z - \partial_\nu a_\mu^z)^2. \tag{50}$$

For simplicity, let us consider a K matrix in the form:

$$K_{z,z'} = \begin{pmatrix} c_0 & c_1 & & & c_1 \\ c_1 & c_0 & c_1 & & \\ & \ddots & \ddots & \ddots & \\ & & c_1 & c_0 & c_1 \\ c_1 & & & c_1 & c_0 \end{pmatrix}, \tag{51}$$

which has translational symmetry along the $z$ direction so we can employ a Fourier transformation to diagonalize it. This theory describes a phase with dispersion relation

$$\omega^2 = \mathbf{q}^2 + C(q_z)^2 g^4, \tag{52}$$

where $C(q_z) = \frac{1}{2\pi}(c_0 + 2c_1 \cos q_z)$, $q_z = 0, \frac{2\pi}{N}, \cdots, \frac{2\pi}{N}(N-1)$ is the Fourier momentum in $z$ direction. To avoid complexity, we take the limit $N \to \infty$ so $q_z$ can be any rational or irrational number $\in [0, 2\pi)$. We can see that when the ratio $|c_0/2c_1| > 1$, $C(q_z)$ is always nonzero and the energy dispersion Eq. 52 has a minimum energy

$$\omega_{\min} = \frac{g^2}{2\pi}(|c_0| - 2|c_1|) \tag{53}$$

which occurs at either $q_z = 0$ or $q_z = \pi$ depending on the relative sign of $c_0$ and $c_1$. In this case, the photon has a large gap. However, with other values of the ratio $c_0/2c_1$, the photon can also be gapless in the 2D iCSM.

When the ratio $|c_0/2c_1| = 1$, $C(q_z) = 0$ at $q_z^*$ which equals to 0 or $\pi$. The photon is gapless with a quadratic dispersion in $q_z$:

$$\omega^2 = \mathbf{q}^2 + \frac{c_0^2 g^4}{16\pi^2}(q_z - q_z^*)^4, \quad |q_z - q_z^*| \ll 1. \tag{54}$$

When the ratio $|c_0/2c_1| < 1$, $C(q_z) = 0$ at $q_z^* = \pm \arccos(-c_0/2c_1)$. The photon is gapless with a linear dispersion in $q_z$:

$$\omega^2 = \mathbf{q}^2 + \frac{\left(4c_1^2 - c_0^2\right)g^4}{4\pi^2}\left(q_z^2 - q_z^*\right)^2, \quad |q_z - q_z^*| \ll 1. \tag{55}$$

Other interesting features of this 2D iCSM are discussed in Ref. [29]. For example, when $c_0/2c_1 > 1$, we can get the inverse of the $K$ matrix when $N \to \infty$:

$$\left(K^{-1}\right)_{z,z'} \to \frac{(-1)^{z-z'}}{2c_1\sqrt{\left(\frac{c_0}{2c_1}\right)^2 - 1}}\left(\frac{c_0}{2c_1} + \sqrt{\left(\frac{c_0}{2c_1}\right)^2 - 1}\right)^{-|z-z'|}. \tag{56}$$

In Ref. [29], the author chose $c_0 = 3, c_1 = 1$. This describes a gapped phase with quite strange statistics $\theta_{z,z'} = 2\pi \frac{(-1)^{z-z'}}{\sqrt{5}}\left(\frac{3+\sqrt{5}}{2}\right)^{-|z-z'|}$, which decay exponentially when $z - z'$ grows, but never become exactly zero. It was called non-foliated fracton order [29].

## 5.2 3D iCSM theory: a gapless phase

The U(1) gauge field in the iCSM theory is still 2+1d. Now we consider a phase transition after which it becomes 3+1d. As before we simply consider the onset of a term $\sum_z \Phi(z)b^\dagger(z)b(z+1)$ where $b(z)$ is the operator carrying charge 1 under the gauge field in the $z$ layer. Then this term makes the U(1) gauge field 3D, similar to our previous discussions on the 3D CSL phase. Let us first understand its property. The action now is:

$$S_{\text{3D,iCSM}} = \frac{i}{4\pi}\sum_{z,z'}K_{z,z'}\int \mathrm{d}^3 x\,\epsilon_{\mu\nu\rho}a_\mu^z\partial_\nu a_\rho^{z'} + \frac{1}{4g^2}\sum_z \int \mathrm{d}^3 x(\partial_\mu a_\nu^z - \partial_\nu a_\mu^z)^2$$
$$+ \frac{\rho_s}{2}\sum_z \int \mathrm{d}^3 x(\partial_\mu a_z - (a_\mu^{z+1} - a_\mu^z))^2. \tag{57}$$

Note that we can use the gauge $a_z = 0$, so the $\rho_s$ term just looks like a Higgs term (see Eq. 25):

$$S_{\text{int}} = \frac{1}{2}\sum_{q_z}\int \frac{\mathrm{d}^3 q}{(2\pi)^3}u(q_z)a_\mu^{q_z}(q)\left(\delta_{\mu\nu} - \frac{q_\mu q_\nu}{q^2}\right)a_\nu^{-q_z}(-q), \tag{58}$$

where $\mu, \nu = 0, 1, 2$, $q_z = 0, \frac{2\pi}{N}, \dots \frac{2\pi(N-1)}{N}$ is the discrete momentum in z-direction, the coefficient $u(q_z) = 4\rho_s \sin^2(q_z/2)$.

In our discussion, we assume that the matrix $K_{z,z'}$ only includes diagonal and the nearest neighbor terms, $K_{z,z'} = c_0\delta_{z,z'} + c_1\delta_{z,z'+1} + c_1\delta_{z,z'-1}$. Fourier transforming the action Eq.(57), we get:

$$S_{\text{3D,iCSM}} = \frac{1}{2}\sum_{q_z}\int \frac{\mathrm{d}^3 q}{(2\pi)^3}a_\mu^{q_z}(q)\left(C(q_z)\epsilon_{\mu\rho\nu}q_\rho + \left(\frac{1}{g^2}q^2 + u(q_z)\right)\left(\delta_{\mu\nu} - \frac{q_\mu q_\nu}{q^2}\right)\right)a_\nu^{-q_z}(-q), \tag{59}$$

where $C(q_z) = \frac{1}{2\pi}(c_0 + 2c_1\cos q_z)$, $u(q_z) = 4\rho_s\sin^2\frac{q_z}{2}$. By doing the matrix inverse in the transverse subspace, we obtain the photon propagator

$$D_{\mu\nu}^{q_z}(q) = \frac{-C}{C^2 q^2 + \left(\frac{q^2}{g^2} + u(q_z)\right)^2}\epsilon_{\mu\nu\rho}q_\rho + \frac{\frac{q^2}{g^2} + u(q_z)}{C^2 q^2 + \left(\frac{q^2}{g^2} + u(q_z)\right)^2}\left(\delta_{\mu\nu} - \frac{q_\mu q_\nu}{q^2}\right). \tag{60}$$

360    From its poles, we can obtain the dispersion relations. We get two modes with dispersion
361 relations

$$\omega_{\pm}^2 = \mathbf{q}^2 + \frac{1}{2}\left(C(q_z)^2 g^4 + 2u(q_z)g^2 \pm \sqrt{C(q_z)^4 g^8 + 4C(q_z)^2 u(q_z)g^6}\right). \tag{61}$$

362    The reason why we get two photon modes here instead of one mode in 2D iCSM is the
363 same as in Sec. 3: after the condensation transition, the phase of the condensate serve as a
364 new gauge field component, so the gauge field is now 3+1 d. We know that there can be two
365 polarizations in 3+1 d $U(1)$ gauge theory. The difference of Eq. 61 from the energy dispersion
366 in Sec. 3 is that there is a $q_z$ dependent function $C(q_z)$ which comes from the inter-layer mutual
367 coupling $K_{z,z'}$. The low energy dispersion relations are summarized in Fig. 7. We can see that
368 in 3D iCSM, the new photon mode $\omega_-$ is always gapless for all values of $c_0/2c_1$. For most
369 situations, $\omega_-$ is quadratic in small $q_z$ while when $c_0/2c_1 = -1$ it is linear in small $q_z$.
370    We also discussed the electromagnetic response of the 3D iCSM theory in Appendix. D.
371 Both $\sigma_{xx}$ and $\sigma_{xy}$ vanish in the DC ($\omega = 0$) limit at finite $q_z \neq 0$, like a trivial insulator. Only
372 at $q_z = 0$, $\sigma_{xx}^{q_z=0}(\omega = 0) = 0$ and $\sigma_{xy}^{q_z=0}(\omega = 0) = \frac{1}{c_0 + 2c_1}\frac{e^2}{h}$, like a FQHE insulator.

### 5.3  Critical theory between gapped fracton order and gapless 3D phase

374 Similar to Eq. 30, we can also write down the critical theory at the transition point $g = g_c(c_0, c_1)$
375 and the only difference from Eq. 30 is the Chern-simons term:

$$\begin{aligned}
S = \sum_z \int \mathrm{d}^3 x \left| \left(\partial_\mu - i\left(a_\mu^{z+1} - a_\mu^z\right)\right)\Phi_z\right|^2 + \frac{i}{4\pi}\sum_{z,z'} K_{z,z'} \int \mathrm{d}^3 x \, \epsilon_{\mu\nu\rho} a_\mu^z \partial_\nu a_\rho^{z'} \\
+ \frac{1}{2}\sum_{z,z'}\lambda_{z,z'}\int \mathrm{d}^3 x |\Phi_z|^2 |\Phi_{z'}|^2,
\end{aligned} \tag{62}$$

376 where $\Phi_z$ is the condensate field between layer $z$ and $z+1$. We can still use the large $N_b$
377 expansion we used in Sec. 4 to study the critical behavior. The only difference is in the bare
378 photon propagator. Note that $|c_0/2c_1| > 1$ is required for the $K$ matrix to be inverted. For
379 example, when $c_0 = 3, c_1 = 1$, We can get the bare photon propagator:

$$D_{0,\mu\nu}^{q_z}(q) = -\frac{2\pi}{N_b}\frac{\epsilon_{\mu\lambda\nu}q_\lambda}{q^2}\frac{1}{3 + 2\cos q_z}, \tag{63}$$

380 or in real space:

$$D_{0,\mu\nu}^{z,z'}(q) = -\frac{2\pi}{N_b}\frac{\epsilon_{\mu\lambda\nu}q_\lambda}{q^2}(K)_{z,z'}^{-1}, \tag{64}$$

381 where from Eq. 56 we have

$$(K)_{z,z'}^{-1} = \frac{(-1)^{z-z'}}{\sqrt{5}}\left(\frac{3+\sqrt{5}}{2}\right)^{-|z-z'|}. \tag{65}$$

382    Unlike Eq. 32, where the bare photon propagator is diagonal in $z, z'$, here the bare photon
383 propagator already has off-diagonal terms. In other words, the bare photon propagator is
384 already $q_z$ dependent in momentum space. We can further calculate the large $N_b$ effective
385 photon propagator

$$D_{\text{eff},\mu\nu}^{q_z}(q) = \frac{2\pi/(3+2\cos q_z)}{1 + \frac{\pi^2 \sin^4\frac{q_z}{2}}{4(3+2\cos q_z)^2}}\left(\frac{\pi\sin^2\frac{q_z}{2}}{2(3+2\cos q_z)|q|}\left(\delta_{\mu\nu} - \frac{q_\mu q_\nu}{q^2}\right) - \frac{\epsilon_{\mu\nu\lambda}q_\lambda}{q^2}\right) + \mathcal{O}(1/N_b^2), \tag{66}$$

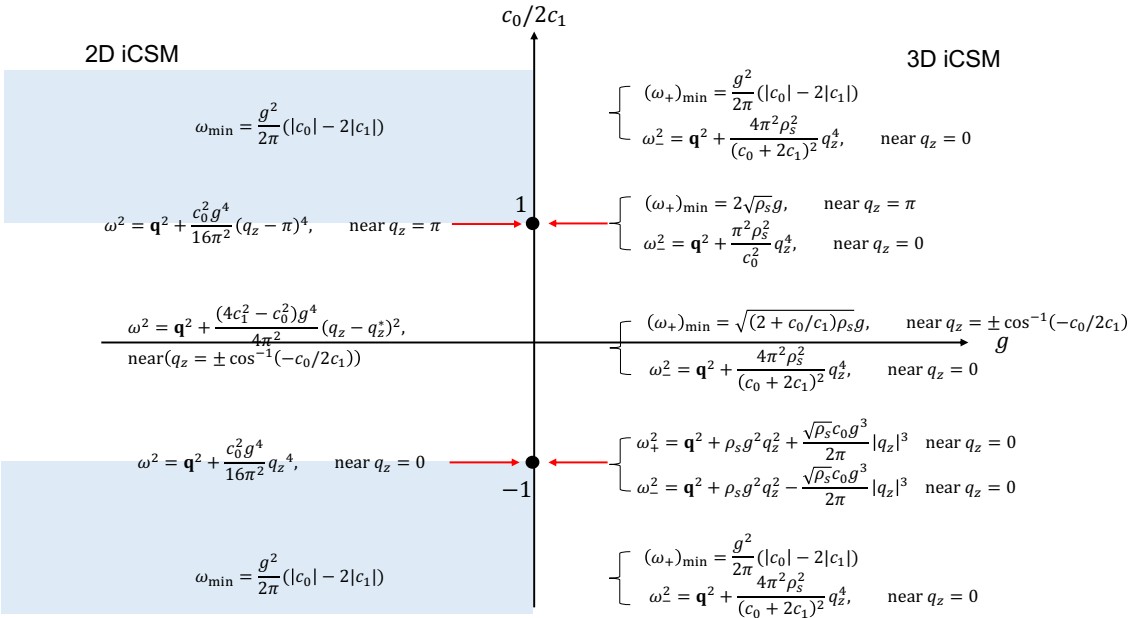

Figure 7: Phase diagram of the iCSM theory. $g$ is the control parameter driving the system go through the transition from 2D iCSM to 3D iCSM. The shaded regions represent the gapped phase while others are the gapless phase. The dispersion relations are not valid near the critical region $g = g_c$, whose properties need more careful studies.

which is also $q_z$ dependent, and see that the photon energy is still zero for any $q_z$ as long as $\mathbf{q} = 0$. This slight difference in the bare photon propagator will not change the overall critical behavior we discussed in Sec. 4.

# 6 Conclusion

In summary, we propose a general framework to understand the 2D to 3D transition of a fractional phase with a U(1) gauge field in a system with infinitely stacked 2D layers. We applied it to the case of chiral spin liquid (or fractional quantum Hall phase). The 3D phase of the chiral spin liquid (CSL) has a gapless photon mode. The 2D to 3D transition is described by the Higgs transition of a boson $\Phi$, which becomes critical at each layer. Interestingly, we find that the critical mode is gapless along a line $(q_x, q_y, q_z) = (0, 0, q_z)$ for any $q_z \in [0, 2\pi)$, but the scaling dimension has a $q_z$ dependence. As a result, gauge invariant operators have a finite but non-zero correlation length in the $z$-direction. Besides, our theory can be generalized to describe a continuous phase transition between a fracton phase described by infinite component Chern Simons theory and a 3D gapless phase similar to the 3D CSL. In the future, we hope to make the matter field also gapless at the critical point. For example, we can generalize the current framework to describe the 2D to 3D transition of composite Fermi liquid, Dirac spin liquid, and spinon Fermi surface phases. It is also interesting to study metal-insulator transition in quasi 2D system [39] with 2D or 3D spin liquid in the insulator side.

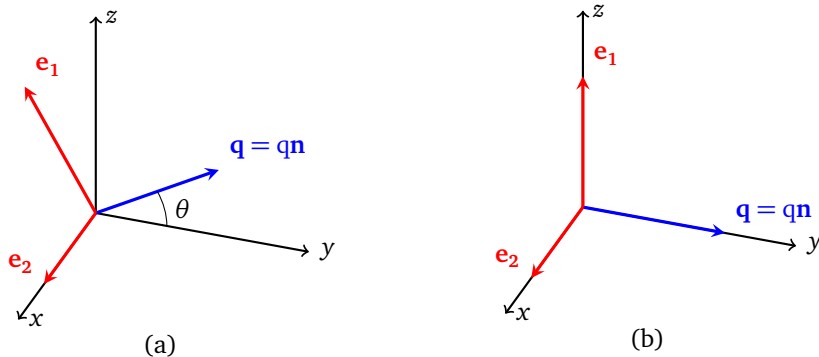

Figure 8: Definition of vectors we use to calculate the photon polarizations. Unit vectors $\mathbf{e_1}$, $\mathbf{e_2}$ and $\mathbf{n}$ form a right-hand orthonormal system. $\mathbf{n}$ lies in the $y$-$z$ plane and represents the direction of wave vector $\mathbf{q}$. $\mathbf{e_2}$ is always pointing in the $x$-direction. (a) is for nonzero $\theta$ case and (b) is for $\theta = 0$ case.

## Acknowledgements

YHZ thanks Ashvin Vishwanath for discussions at initial stage of the work. This work was supported by the National Science Foundation under Grant No. DMR2237031. This work was performed in part at Aspen Center for Physics, which is supported by National Science Foundation grant PHY-2210452.

## A  Plane-wave solution to the Maxwell equations

The new set of "Maxwell's equations" in the 3D CSL phase is as follows:

$$\nabla \cdot \vec{e} + (\tilde{\rho}_s g^2 - 1)\partial_z e_z - \frac{kg^2}{2\pi} b_z = 0, \tag{A.1}$$

$$\partial_t \vec{e} - \nabla \times \vec{b} - (\tilde{\rho}_s g^2 - 1)\partial_z(\hat{z} \times \vec{b}) - \frac{kg^2}{2\pi}\hat{z} \times \vec{e} = 0, \tag{A.2}$$

$$\nabla \cdot \vec{b} = 0, \tag{A.3}$$

$$\nabla \times \vec{e} + \partial_t \vec{b} = 0. \tag{A.4}$$

To find the plane-wave solution $\vec{e} = \mathcal{E}e^{i\mathbf{q}\cdot\mathbf{x} - i\omega t}$, $\vec{b} = \mathcal{B}e^{i\mathbf{q}\cdot\mathbf{x} - i\omega t}$ to the equations above, we set $\mathbf{q}$ lying in the $y$-$z$ plane due to the rotational symmetry about the $z$-axis. Eq. A.3 is then $\mathbf{n} \cdot \mathcal{B} = 0$, so $\mathcal{B} = B_1 \mathbf{e_1} + B_2 \mathbf{e_2}$ where $\mathbf{e_1} = (0, -\sin\theta, \cos\theta)$, $\mathbf{e_2} = (1, 0, 0)$. Using Eq. A.1 and Eq. A.4, we get $\mathcal{E} = \frac{\omega B_2}{q}\mathbf{e_1} - \frac{\omega B_1}{q}\mathbf{e_2} + \frac{\omega B_2 \sin\theta\cos\theta(1 - \tilde{\rho}_s g^2) - \frac{ikg^2}{2\pi}B_1 \cos\theta}{q(\cos^2\theta + \tilde{\rho}_s g^2 \sin^2\theta)}\mathbf{n}$. Plug them into Eq. A.2, we get

$$\begin{pmatrix} -\frac{kg^2\omega\sin\theta}{2\pi} & -i(\omega^2 - q^2) + i(\tilde{\rho}_s g^2 - 1)q_z^2 \\ i(\omega^2 - q^2) - i(\tilde{\rho}_s g^2 - 1)q_z^2 - \frac{ik^2 g^4 \cos^2\theta}{4\pi^2(\cos^2\theta + \tilde{\rho}_s g^2 \sin^2\theta)} & -\frac{k\tilde{\rho}_s g^4 \omega\sin\theta}{2\pi(\cos^2\theta + \tilde{\rho}_s g^2 \sin^2\theta)} \end{pmatrix}\begin{pmatrix} B_1 \\ B_2 \end{pmatrix} = 0. \tag{A.5}$$

For plane wave solutions to exist, the determinant of the large matrix should vanish. Solving this equation gives us two dispersion relations $\omega_\pm$ and the corresponding two eigenmodes $\mathcal{B}_\pm$:

$$\omega_\pm^2 = q_x^2 + q_y^2 + \tilde{\rho}_s g^2 q_z^2 + \frac{k^2 g^4}{8\pi^2} \pm \frac{k^2 g^4}{8\pi^2}\sqrt{1 + \frac{16\pi^2 \tilde{\rho}_s}{k^2 g^2} q_z^2} \tag{A.6}$$

$$\mathcal{B}_+ = \mathbf{e_1} + \frac{ik\left(-\cos^2\theta + \tilde{\rho}_s g^2 \sin^2\theta + (\cos^2\theta + \tilde{\rho}_s g^2 \sin^2\theta)\sqrt{1 + \frac{16\pi^2 \tilde{\rho}_s}{k^2 g^2} q_z^2}\right)}{\sqrt{2}\tilde{\rho}_s \sin\theta \sqrt{8\pi^2 q^2(\cos^2\theta + \tilde{\rho}_s g^2 \sin^2\theta) + k^2 g^4 + k^2 g^4\sqrt{1 + \frac{16\pi^2 \tilde{\rho}_s}{k^2 g^2} q_z^2}}} \mathbf{e_2}, \tag{A.7}$$

$$\mathcal{B}_- = \frac{i\sqrt{2}\tilde{\rho}_s \sin\theta \sqrt{8\pi^2 q^2(\cos^2\theta + \tilde{\rho}_s g^2 \sin^2\theta) + k^2 g^4 - k^2 g^4\sqrt{1 + \frac{16\pi^2 \tilde{\rho}_s}{k^2 g^2} q_z^2}}}{k\left(\cos^2\theta - \tilde{\rho}_s g^2 \sin^2\theta + (\cos^2\theta + \tilde{\rho}_s g^2 \sin^2\theta)\sqrt{1 + \frac{16\pi^2 \tilde{\rho}_s}{k^2 g^2} q_z^2}\right)} \mathbf{e_1} + \mathbf{e_2}. \tag{A.8}$$

We can look at the special case where $\theta = 0$ (see Fig. 8). Here $\mathcal{B}_\pm$ both become linearly polarized: $\mathcal{B}_+ = \mathbf{e_1}$ and $\mathcal{B}_- = \mathbf{e_2}$. Remember that in 2+1 d Chern-Simons theory, there is only one gapped mode and the magnetic field only has a z-component. Here $\mathcal{B}_+$ looks very similar to that mode: it is linearly polarized in the z-direction and has a large energy gap which goes to infinity as $g^2 \to \infty$ as in $2 + 1$ d Chern-Simons theory. The other gapless mode $\mathcal{B}_-$ lies in the x-y plane, which cannot exist unless we have the fourth component of the gauge field. So starting from the 2D CSL, the gapped photon mode remains gapped across the transition, while a new gapless mode emerges after the transition.

# B  Effective propagators

We consider the effective photon propagator first. In the large $N_b$ limit, all the bubble diagrams (see Fig. 3) are of order unity so we should add them together (a boson loop has factor $N_b$ and a bare propagator has factor $1/N_b$). Given the definition $\langle \alpha_\mu^z(q)\alpha_\nu^{z'}(-q')\rangle = (2\pi)^3 \delta^3(q-q')\tilde{D}_{\mu\nu}^{z-z'}(q)$, we have a Dyson equation

$$\tilde{D}_0(q) + \tilde{D}_0(q)\tilde{\Pi}(q)\tilde{D}_{\text{eff}}(q) = \tilde{D}_{\text{eff}}(q), \tag{B.1}$$

where the layer and space-time indices are ignored. The self-energy is

$$\tilde{\Pi}_{\mu\nu}^{z,z'}(q) = N_b \delta^{z,z'} \int \frac{d^3p}{(2\pi)^3} \frac{(2p+q)_\mu (2p+q)_\nu}{(q+p)^2 p^2} = -\frac{N_b}{16}\delta^{z,z'}|q|\left(\delta_{\mu\nu} - \frac{q_\mu q_\nu}{q^2}\right). \tag{B.2}$$

The solution to this equation is $\tilde{D}_{\text{eff}}(q) = (\mathbf{1}(q)-\tilde{D}_0(q)\tilde{\Pi}(q))^{-1}\tilde{D}_0(q)$, where $\mathbf{1}(q) = \delta^{z,z'}\left(\delta_{\mu\nu} - \frac{q_\mu q_\nu}{q^2}\right)$ is the projection operator into transverse subspace since we work in Landau gauge. The matrix inverse is also done in the transverse subspace. It is convenient to go to $q_z$ space and the result is

$$\tilde{D}_{\text{eff},\mu\nu}^{q_z}(q) = \frac{A(q_z)}{N_b}\left(\frac{B(q_z)}{|q|}(\delta_{\mu\nu} - \frac{q_\mu q_\nu}{q^2}) - \frac{\epsilon_{\mu\nu\lambda}q_\lambda}{q^2}\right) + \mathcal{O}(1/N_b^2), \tag{B.3}$$

where we introduced

$$A(q_z) = \frac{\frac{8\pi}{\alpha}\sin^2\frac{q_z}{2}}{1 + \frac{\pi^2 \sin^4\frac{q_z}{2}}{4\alpha^2}}, \quad B(q_z) = \frac{\pi\sin^2\frac{q_z}{2}}{2\alpha}. \tag{B.4}$$

In real $x$ space it is

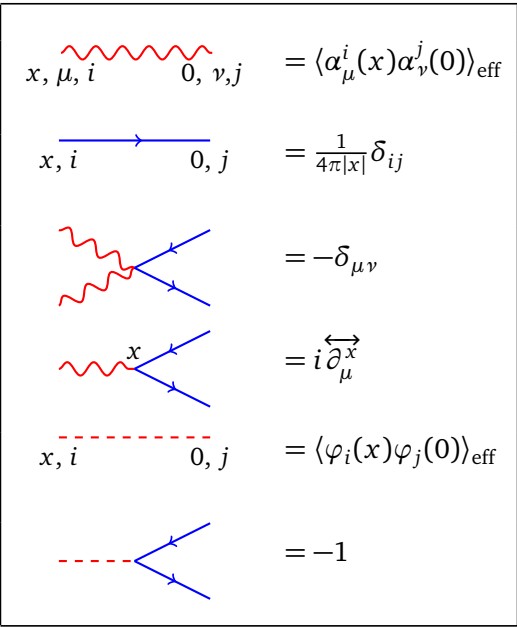

Table 3: Feynman rules. All propagators are diagonal in the flavor index; All vertices conserve the flavor index.

$$\tilde{D}_{\text{eff},\mu\nu}^{q_z}(x) = \frac{A(q_z)}{4\pi^2 N_b}\left(\frac{4B(q_z)x_\mu x_\nu}{|x|^4} + \frac{i\pi\epsilon_{\mu\nu\lambda}x_\lambda}{|x|^3}\right). \tag{B.5}$$

The effective propagator $G_{\varphi,\text{eff}}$ is also given by bubble diagram summation. Given the definition $\langle\varphi_z(p)\varphi_{z'}(-p')\rangle = (2\pi)^3\delta^3(p-p')G_\varphi^{ij}(p)$, we have a Dyson equation

$$G_{\varphi,0}(p) + G_{\varphi,0}(p)\Pi_\varphi(p)G_{\varphi,\text{eff}}(p) = G_{\varphi,\text{eff}}(p), \tag{B.6}$$

where $G_{\varphi,0}^{z,z'} = -\lambda_{z,z'}$. The self-energy $\Pi_\varphi^{z,z'}(p) = -N_b\delta^{z,z'}\int\frac{d^3q}{(2\pi)^3}\frac{1}{(q+p)^2q^2} = -N_b\frac{\delta^{z,z'}}{8|p|}$. The solution to this equation is $G_{\varphi,\text{eff}}(p) = (\mathbf{1} - G_{\varphi,0}(p)\Pi_\varphi(p))^{-1}G_{\varphi,0}(p)$. The result is

$$G_{\varphi,\text{eff}}^{q_z} = -\frac{8|p|}{N_b} + \mathcal{O}(1/N_b^2). \tag{B.7}$$

In real $x$ and $z$ space, it is

$$G_{\varphi,\text{eff}}^{z-z'}(x) = \frac{8}{\pi^2 N_b|x|^4}\delta^{z,z'}. \tag{B.8}$$

# C   Feynman diagram calculation

All the diagrams and their results are in Table. 2, and the Feynman rules are in Table. 3. We also show the calculation of several typical diagrams in Table. 2. Notice that the result of an individual diagram might depend on the gauge choosing, but the sum of them should not since $\varphi$ is a gauge-independent operator. We use a UV cutoff $\Lambda$ to regularize the divergent momentum integrals. We only keep the logarithmic divergence, so for every $n \neq 3$, $\int\frac{d^3q}{q^n}$ is regarded as 0. To avoid confusion with the integral variable, we use $i, j$ as the layer index in this section.

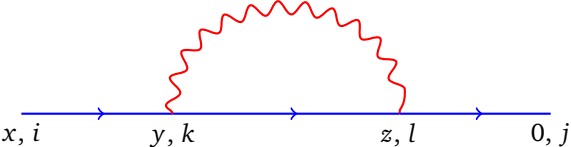

Figure 9: Subdiagram B′: loop correction to the boson propagator

452     First, we consider Graph B. Comparing to Graph A, there is a boson loop giving factor $N_b$,
453  a photon propagator giving factor $1/N_b$ and an extra $G_{\varphi,\text{eff}}$ giving factor $1/N_b$ so the overall
454  order is of $1/N_b$. The two $G_{\varphi,\text{eff}}$'s near the ends don't contribute to logarithmic divergence
455  since in momentum representation, they are just multiplying factors $1/p^2$. Therefore we only
456  need to calculate the logarithmic divergence in the region surrounded by the boson loop (this
457  applies to all the diagrams in Table. 2). It turns out we can calculate the subdiagram B′ in
458  Fig. 9.

$$
\begin{aligned}
\text{Graph B}' &= \sum_k \sum_l \int d^3y\, d^3z \left(\frac{\delta_{ik}}{4\pi|x-y|}\right) i\overset{\leftrightarrow}{\partial}{}^{\,y}_\mu \left(\frac{\delta_{kl}}{4\pi|y-z|}\right) i\overset{\leftrightarrow}{\partial}{}^{\,z}_\nu \left(\frac{\delta_{lj}}{4\pi|z|}\right) \cdot \tilde{D}^{kl}_{\text{eff},\mu\nu}(y-z) \\
&= \delta_{ij} \int d^3y\, d^3z \left(\frac{1}{4\pi|x-y|}\right) i\overset{\leftrightarrow}{\partial}{}^{\,y}_\mu \left(\frac{1}{4\pi|y-z|}\right) i\overset{\leftrightarrow}{\partial}{}^{\,z}_\nu \left(\frac{1}{4\pi|z|}\right) \cdot \frac{1}{N}\sum_{q_z} \tilde{D}^{q_z}_{\text{eff},\mu\nu}(y-z) \\
&= \delta_{ij} \int \frac{d^3p}{(2\pi)^3} \frac{e^{ip\cdot x}}{p^4} \left[\frac{1}{N}\sum_{q_z}\int \frac{d^3q}{(2\pi)^3} \frac{(2p+q)_\mu(2p+q)_\nu}{(q+p)^2} \tilde{D}^{q_z}_{\text{eff},\mu\nu}(-q)\right] \\
&= \delta_{ij} \int \frac{d^3p}{(2\pi)^3} \frac{e^{ip\cdot x}}{p^4} \left[\frac{1}{N}\sum_{q_z}\int \frac{d^3q}{(2\pi)^3} \frac{(2p+q)_\mu(2p+q)_\nu}{(q+p)^2} \right. \\
&\qquad \left. \times \frac{A(q_z)}{N_b}\left(\frac{B(q_z)}{|q|}(\delta_{\mu\nu} - \frac{q_\mu q_\nu}{q^2}) - \frac{\epsilon_{\mu\nu\lambda}q_\lambda}{q^2}\right)\right].
\end{aligned}
\tag{C.1}
$$

459  We do the q integral first. Using $\frac{1}{(p+q)^2} = \frac{1}{q^2} - \frac{2q\cdot p + p^2}{q^4} + \frac{4(p\cdot q)^2}{q^6} + \mathcal{O}(\frac{1}{q^6})$, we obtain its logarithmic
460  divergence to be

$$
\begin{aligned}
\text{Graph B}' &= \delta_{ij} \int \frac{d^3p}{(2\pi)^3} \frac{e^{ip\cdot x}}{p^4} \left[\frac{1}{N_bN}\sum_{q_z} A(q_z)B(q_z) \int \frac{d^3q}{(2\pi)^3} \frac{4p_\mu p_\nu}{|q|^3}(\delta_{\mu\nu} - \frac{q_\mu q_\nu}{q^2})\right] \\
&= \delta_{ij} \int \frac{d^3p}{(2\pi)^3} \frac{e^{ip\cdot x}}{p^4} \left[\frac{1}{N_bN}\sum_{q_z} A(q_z)B(q_z) \cdot 4p_\mu p_\nu \delta_{\mu\nu}(1 - \frac{1}{3}) \int \frac{d^3q}{(2\pi)^3} \frac{1}{|q|^3}\right],
\end{aligned}
\tag{C.2}
$$

461  where we used $\int d^dq\, f(q^2) q_\mu q_\nu = \frac{1}{d}\int d^dq\, q^2 f(q^2)\delta_{\mu\nu}$. The last q integral is regularized by
462  cutoff momentum $\Lambda$:

$$
\int \frac{d^3q}{(2\pi)^3} \frac{1}{|q|^3} = \frac{1}{4\pi^2}\ln x^2\Lambda^2.
\tag{C.3}
$$

463  Here we use the dimensionless parameter $x\Lambda$ inside the logarithmic function. So,

$$
\begin{aligned}
\text{Graph B}' &= \delta_{ij} \cdot \frac{2}{3\pi^2 N_bN} \sum_{q_z} A(q_z)B(q_z)\ln x^2\Lambda^2 \cdot \int \frac{d^3p}{(2\pi)^3} \frac{e^{ip\cdot x}}{p^2} \\
&= \delta_{ij} \cdot \frac{2}{3\pi^2 N} \sum_{q_z} A(q_z)B(q_z)\ln x^2\Lambda^2 \cdot \left(\frac{1}{4\pi|x|}\right).
\end{aligned}
\tag{C.4}
$$

464    Then using

$$\frac{1}{|x|^{2\alpha}} = \frac{\Gamma(\frac{d}{2} - \alpha)}{\pi^{\frac{d}{2}} 2^{2\alpha} \Gamma(\alpha)} \int d^d p \, \frac{e^{ip\cdot x}}{|p|^{d-2\alpha}}, \tag{C.5}$$

465    we have

$$\int d^3y \, d^3z \left( \frac{8}{\pi^2 N_b |x-y|^4} \right) \left( \frac{1}{4\pi |y-z|} \right)^2 \left( \frac{8}{\pi^2 N_b |z|^4} \right) = -\frac{8}{\pi^2 N_b |x|^4}, \tag{C.6}$$

466    so we get the result of Graph B in Table. 2.

467       Next, we consider the Graph F. In Graph F, there are two boson loops that contribute a factor
468    $N_b^2$, two photon propagators that contribute a factor $1/N_b^2$, and an extra $G_{\varphi,\text{eff}}$ that contribute
469    a factor $1/N_b$. So this diagram is of order $1/N_b$ compared to Graph A. Again we calculate the
470    amputated subdiagram without two external $G_{\varphi,\text{eff}}$ first,

Graph F (amputated)

$$= 4N_b^2 \int d^3y \, d^3z \, d^3w \left( \frac{1}{4\pi|x-y|} \right)^2 (-\delta_{\mu\nu}) \tilde{D}_{\text{eff},\mu\alpha}^{ij}(y-z) \tilde{D}_{\text{eff},\nu\beta}^{ij}(y-w)$$
$$\times \left[ \frac{1}{4\pi|w|} i \overset{\leftrightarrow}{\partial}{}_\beta^w \frac{1}{4\pi|w-z|} i \overset{\leftrightarrow}{\partial}{}_\alpha^z \frac{1}{4\pi|z|} \right]. \tag{C.7}$$

471    Since $\tilde{D}_{\text{eff}}(x) \propto 1/x^2$, by power counting, the $\ln \Lambda$ divergence might come from the region
472    where $y, z, w \to x$ or the region where $y, z, w \to 0$. In the first region, the logarithmic divergent
473    part is

Region 1

$$= -\frac{4N_b^2}{(4\pi|x|)^2} \int d^3y \, d^3z \, d^3w \left( \frac{1}{4\pi|x-y|} \right)^2 \tilde{D}_{\text{eff},\mu\alpha}^{ij}(y-z) \tilde{D}_{\text{eff},\mu\beta}^{ij}(y-w) \cdot \left( \overset{\rightarrow}{\partial}{}_\beta^w \overset{\rightarrow}{\partial}{}_\alpha^z \frac{1}{4\pi|w-z|} \right)$$
$$= -\frac{4N_b^2}{(4\pi|x|)^2} \int d^3y' \, d^3z' \, d^3w' \left( \frac{1}{4\pi|y'|} \right)^2 \tilde{D}_{\text{eff},\mu\alpha}^{ij}(-z') \tilde{D}_{\text{eff},\mu\beta}^{ij}(-w') \cdot \left( \overset{\rightarrow}{\partial}{}_\beta^{w'} \overset{\rightarrow}{\partial}{}_\alpha^{z'} \frac{1}{4\pi|w'-z'|} \right) \tag{C.8}$$

474    where we introduced new integral variables $y' = y - x$, $z' = z - y$, $w' = w - y$. The
475    integral over $y'$ has no UV divergence ($y' \to 0$); the remaining integral by power counting
476    should be proportional to $\int d^3x / x^4$, which is not a logarithmic divergence.

477       In the second region where $y, z, w \to 0$, the logarithmic divergent part is

Region 2

$$= -\frac{4N_b^2}{(4\pi|x|)^2} \int d^3y \, d^3z \, d^3w \, \tilde{D}_{\text{eff},\mu\alpha}^{ij}(y-z) \tilde{D}_{\text{eff},\mu\beta}^{ij}(y-w) \left[ \frac{1}{4\pi|w|} i \overset{\leftrightarrow}{\partial}{}_\beta^w \frac{1}{4\pi|w-z|} i \overset{\leftrightarrow}{\partial}{}_\alpha^z \frac{1}{4\pi|z|} \right]$$
$$= -\frac{4N_b^2}{(4\pi|x|)^2} \frac{1}{N^2} \sum_{q_z, l_z} e^{iq_z \cdot (i-j)} \int \frac{d^3p}{(2\pi)^3} \frac{d^3q}{(2\pi)^3} \tilde{D}_{\text{eff},\mu\alpha}^{l_z}(p) \tilde{D}_{\text{eff},\mu\beta}^{q_z - l_z}(-p) \frac{(p+2q)_\alpha (p+2q)_\beta}{(p+q)^4 q^2}$$
$$= -\frac{4}{(4\pi|x|)^2} \frac{1}{N^2} \sum_{q_z, l_z} e^{iq_z \cdot (i-j)} A(l_z) A(q_z - l_z) \int \frac{d^3p}{(2\pi)^3} \frac{d^3q}{(2\pi)^3} \frac{(p+2q)_\alpha (p+2q)_\beta}{(p+q)^4 q^2}$$
$$\times \left[ \frac{B(l_z)}{|p|} (\delta_{\mu\alpha} - \frac{p_\mu p_\alpha}{p^2}) - \frac{\epsilon_{\mu\alpha\lambda} p_\lambda}{p^2} \right] \left[ \frac{B(q_z - l_z)}{|p|} (\delta_{\mu\beta} - \frac{p_\mu p_\beta}{p^2}) + \frac{\epsilon_{\mu\beta\sigma} p_\sigma}{p^2} \right]. \tag{C.9}$$

478       We calculate the 4 crossing terms one by one. The first term has an integral

Integral 1

$$
= \int \frac{d^3p}{(2\pi)^3} \frac{d^3q}{(2\pi)^3} \frac{(p+2q)_\alpha (p+2q)_\beta}{p^2(p+q)^4 q^2} \cdot (\delta_{\mu\alpha} - \frac{p_\mu p_\alpha}{p^2})(\delta_{\mu\beta} - \frac{p_\mu p_\beta}{p^2})
$$

$$
= \int \frac{d^3p}{(2\pi)^3} \frac{d^3q}{(2\pi)^3} \left( \frac{(p+2q)^2}{p^2(p+q)^4 q^2} - \frac{(p\cdot(p+2q))^2}{p^4(p+q)^4 q^2} \right)
$$

$$
= \text{Integral 1.1} - \text{Integral 1.2}.
$$

(C.10)

First, we show that Integral 1.2 vanishes:

Integral 1.2

$$
= \int \frac{d^3p}{(2\pi)^3} \frac{d^3q}{(2\pi)^3} \frac{(p^2 + 2p\cdot q)^2}{p^4(p+q)^4 q^2}
$$

$$
= \int \frac{d^3p}{(2\pi)^3} \frac{d^3q}{(2\pi)^3} \frac{\left((p+q)^2 - q^2\right)^2}{p^4(p+q)^4 q^2}
$$

$$
= \int \frac{d^3p}{(2\pi)^3} \frac{d^3q}{(2\pi)^3} \left( \frac{1}{p^4 q^2} + \frac{q^2}{p^4(p+q)^4} - \frac{2}{p^4(p+q)^2} \right)
$$

$$
= \int \frac{d^3p}{(2\pi)^3} \frac{d^3q}{(2\pi)^3} \left( \frac{1}{p^4 q^2} + \frac{(q-p)^2}{p^4 q^4} - \frac{2}{p^4 q^2} \right)
$$

$$
= \int \frac{d^3p}{(2\pi)^3} \frac{d^3q}{(2\pi)^3} \left( -\frac{2p\cdot q}{p^4 q^2} + \frac{1}{p^2 q^4} \right)
$$

$$
= 0.
$$

(C.11)

In the intermediate steps we shifted the integral variables. Then we calculate Integral 1.1:

Integral 1.1

$$
= \int \frac{d^3p}{(2\pi)^3} \frac{d^3q}{(2\pi)^3} \frac{(p+2q)^2}{p^2(p+q)^4 q^2}
$$

$$
= \int \frac{d^3p}{(2\pi)^3} \frac{d^3q}{(2\pi)^3} \frac{(p+q)^2}{p^4(p-q)^2 q^2}
$$

$$
= \int \frac{d^3p}{(2\pi)^3} \frac{d^3q}{(2\pi)^3} \frac{(p-q)^2 + 4p\cdot q}{p^4(p-q)^2 q^2}
$$

$$
= \int \frac{d^3p}{(2\pi)^3} \frac{d^3q}{(2\pi)^3} \frac{4p\cdot q}{p^4(p-q)^2 q^2}.
$$

(C.12)

Here we use some useful identities below, which can be derived with the help of Feynman parametrization:

$$
\int \frac{d^3q}{(2\pi)^3} \frac{1}{q^2(q+p)^2} = \frac{1}{8|p|},
$$

(C.13)

$$
\int \frac{d^3q}{(2\pi)^3} \frac{q_\mu}{q^4(q+p)^2} = -\frac{p_\mu}{16|p|^3}.
$$

(C.14)

Then

Integral 1.1

$$
\begin{aligned}
&= \int \frac{d^3q}{(2\pi)^3} 4q_\mu \int \frac{d^3p}{(2\pi)^3} \frac{p_\mu}{p^4(p-q)^2} \\
&= \int \frac{d^3q}{(2\pi)^3} \frac{4q_\mu}{q^2} \frac{q_\mu}{16|q|^3} \\
&= \int \frac{d^3q}{(2\pi)^3} \frac{1}{4|q|^3} \\
&= \frac{1}{16\pi^2} \ln x^2\Lambda^2,
\end{aligned} \tag{C.15}
$$

thus we finish the calculation of the first integral and get

$$
\text{Integral 1} = \frac{1}{16\pi^2} \ln x^2\Lambda^2. \tag{C.16}
$$

The second and third integral vanish. For example,

$$
\text{Integral 2} = \int \frac{d^3p}{(2\pi)^3} \frac{d^3q}{(2\pi)^3} \frac{(p+2q)_\alpha(p+2q)_\beta}{|p|^3(p+q)^4q^2}(\delta_{\mu\alpha} - \frac{p_\mu p_\alpha}{p^2})\epsilon_{\mu\beta\sigma}p_\sigma = 0 \tag{C.17}
$$

vanishes due to the anti-symmetric symbol.

Finally, we do the fourth integral. Using $\epsilon_{\mu\alpha\lambda}\epsilon_{\mu\beta\sigma} = \delta_{\alpha\beta}\delta_{\lambda\sigma} - \delta_{\alpha\sigma}\delta_{\lambda\beta}$,

Integral 4

$$
\begin{aligned}
&= \int \frac{d^3p}{(2\pi)^3} \frac{d^3q}{(2\pi)^3} \frac{-(p+2q)_\alpha(p+2q)_\beta}{p^4(p+q)^4q^2}\epsilon_{\mu\alpha\lambda}\epsilon_{\mu\beta\sigma}p_\lambda p_\sigma \\
&= \int \frac{d^3p}{(2\pi)^3} \frac{d^3q}{(2\pi)^3} \frac{(p\cdot(p+2q))^2 - p^2(p+2q)^2}{p^4(p+q)^4q^2} \\
&= \int \frac{d^3p}{(2\pi)^3} \frac{d^3q}{(2\pi)^3} \frac{[(p+q)^2 - q^2]^2 - p^2(p+2q)^2}{p^4(p+q)^4q^2} \\
&= \int \frac{d^3p}{(2\pi)^3} \frac{d^3q}{(2\pi)^3} \frac{(p+q)^4 - 2q^2(p+q)^2 + q^4 - p^2(p+2q)^2}{p^4(p+q)^4q^2} \\
&= \int \frac{d^3p}{(2\pi)^3} \frac{d^3q}{(2\pi)^3} \left(\frac{1}{p^4q^2} - \frac{2}{p^4(p+q)^2} + \frac{q^2}{p^4(p+q)^4}\right) - \text{Integral 1.1} \\
&= \int \frac{d^3p}{(2\pi)^3} \frac{d^3q}{(2\pi)^3} \left(\frac{1}{p^4q^2} - \frac{2}{p^4q^2} + \frac{(q-p)^2}{p^4q^4}\right) - \text{Integral 1.1} \\
&= -\text{Integral 1.1} = -\frac{1}{16\pi^2} \ln x^2\Lambda^2.
\end{aligned} \tag{C.18}
$$

So we have

Graph F(amputated)

$$
= -\left(\frac{1}{4\pi|x|}\right)^2 \frac{1}{4\pi^2N^2} \sum_{q_z,l_z} e^{iq_z\cdot(i-j)} A(l_z)A(q_z-l_z)(B(l_z)B(q_z-l_z)-1)\ln x^2\Lambda^2. \tag{C.19}
$$

Using Eq. C.6, we get the result of Graph F in Table. 2.

## D   Electromagnetic response of the iCSM

To calculate its electromagnetic response, we couple the physical current with external electromagnetic field $A_\mu^z$ by adding the following term to Eq. 57:

$$S_c = -\frac{i}{2\pi} \sum_z \int \mathrm{d}^3 x \, \epsilon_{\mu\nu\rho} A_\mu^z \partial_\nu a_\rho^z, \tag{D.1}$$

where we assume unit $U(1)$ charges of the quasiparticle for every layer. Integrating out $a_\mu$, we get the effective action

$$S_{\mathrm{eff}}[A_\mu] = \frac{1}{2} \sum_{q_z} \int \frac{\mathrm{d}^3 q}{(2\pi)^3} A_\mu^{q_z}(q) \Pi_{\mu\nu}^{q_z} A_\nu^{-q_z}(-q), \tag{D.2}$$

with the response kernel

$$\Pi_{\mu\nu}^{q_z}(q) = \frac{1}{4\pi^2} \frac{\left(\frac{q^2}{g^2} + u(q_z)\right)\left(q^2 \delta_{\mu\nu} - q_\mu q_\nu\right)}{C^2 q^2 + \left(\frac{q^2}{g^2} + u(q_z)\right)^2} + \frac{1}{4\pi^2} \frac{C q^2 \epsilon_{\mu\rho\nu} q_\rho}{C^2 q^2 + \left(\frac{q^2}{g^2} + u(q_z)\right)^2}, \tag{D.3}$$

where $C(q_z) = \frac{1}{2\pi}(c_0 + 2c_1 \cos q_z)$, $u(q_z) = 4\rho_s \sin^2 \frac{q_z}{2}$, $\epsilon_{012} = -1$. The conductivity tensor can be obtained from this response kernel by the following equation:

$$\sigma_{ij}^{q_z}(\omega) = \frac{-1}{i\omega} \Pi_{ij}^{q_z}(i\omega \to (\omega + i0^+), \mathbf{q} = 0), \tag{D.4}$$

and we get:

$$\begin{aligned}
\sigma_{xx}^{q_z}(\omega) &= \frac{1}{4\pi^2} \frac{-i\omega(u(q_z) - \omega^2/g^2)}{(u(q_z) - \omega^2/g^2)^2 - C^2\omega^2}, \\
\sigma_{xy}^{q_z}(\omega) &= \frac{1}{4\pi^2} \frac{-C\omega^2}{(u(q_z) - \omega^2/g^2)^2 - C^2\omega^2}.
\end{aligned} \tag{D.5}$$

Note that both $\sigma_{xx}$ and $\sigma_{xy}$ vanish in the DC ($\omega = 0$) limit at finite $q_z \neq 0$, like a trivial insulator. Only at $q_z = 0$, $\sigma_{xx}^{q_z=0}(\omega = 0) = 0$ and $\sigma_{xy}^{q_z=0}(\omega = 0) = \frac{1}{c_0 + 2c_1} \frac{e^2}{h}$, like a FQHE insulator. If we let $\rho_s = 0$ and take the $g^2 \to \infty$ limit, which corresponds to the 2D iCSM, from Eq. D.5 we get the DC conductivity tensor $\sigma_{xx}^{q_z}(\omega = 0) = 0$, $\sigma_{xy}^{q_z}(\omega = 0) = \frac{1}{2\pi(c_0 + 2c_1 \cos q_z)} = \frac{1}{c_0 + 2c_1 \cos q_z} \frac{e^2}{h}$. By Fourier transforming it into real space, we have the following conductivity tensor for 2D iCSM:

$$\begin{aligned}
\sigma_{xx}^{z-z'}(\omega = 0) &= 0, \\
\sigma_{xy}^{z-z'}(\omega = 0) &= \frac{e^2}{h} (K)_{z,z'}^{-1}.
\end{aligned} \tag{D.6}$$

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
