# Peer review of "Two-dimension to three-dimension transition of chiral spin liquid and fractional quantum Hall phases"

_SciPost Physics_

## Round 1 · Referee Report · Anonymous (Referee 1) · 2024-9-30

Report

In this paper, authors study a transition between a phase which corresponds to an infinite number of decoupled layers, with each layer supporting a 2d topological order (authors primarily focus on chiral topological order), and a genuine 3+1-d phase where these layers are strongly coupled together due to the condensation of an inter-layer exciton. The basic idea is that the condensation of the exciton higgses infinitely many gauge fields (one for each layer) simultaneously down to a single (diagonal) gauge field. This 3d phase supports a gapless photon with dispersion that goes like \omega ~ \sqrt(q^2_x + q^2_y + q^4_z). The authors also study the universal aspects of the associated phase transition using a 1/N expansion and at the leading order in 1/N find that the scalar field’s correlation functions decay as a power-law along the layer directions, and exponentially perpendicular to the layers (it shows “local quantum criticality” in the sense that green’s fn has a branch cut at \omega = |k_{in plane}| = 0 for any value of k_z). The authors discuss two different classes of transitions: one where the coupled theory is described by a K-matrix that is diagonal in the layer-index, and the other where it has non-diagonal elements; the latter theory was recently discussed in other papers and referred to as “infinite component Chern-Simons-Maxwell theory”.

Overall, the paper is well written and discusses transitions that have not been considered before. It also provides a different perspective on the infinite component C-S-M theories). The paper is appropriate for publication in SciPost. I do have a few questions/suggestions that authors may want to consider:

  1. In the 3d exciton-condensed phase, the photon correlation functions perpendicular to the layer direction are power-law (since the coefficient of q^4_z term is non-zero), while at the critical point they are exponential. I guess this implies that the coefficient of q^4_z term vanishes as one approaches the transition from the exciton-condensed side? (it seems that this coefficient is proportional to the square of the exciton order parameter). If that's indeed true, it would be nice if authors can comment on the corresponding critical exponents since they capture an important aspect of the transition (namely, the change of the dimensionality).

  2. Gapless phases can have instabilities. For example, a 3d photon phase may have monopoles which may be relevant. I didn’t see any discussion of the monopoles in the 3d phase. Are they irrelevant, or perhaps not allowed for some reason?

  3. The authors say that “it is a good guess to assume \alpha = 2 in this action” while discussing the coefficient to the C-S term in the critical theory. I was wondering if this value is simply forced on us if we are to recover \alpha = 2 in the non-condensed phase? (since gapping out scalars/bosons can’t change the coefficient of the C-S term).

  4. The authors rely crucially on translation symmetry along the z-direction to accomplish the Higgsing of infinitely many gauge fields simultaneously. Have authors thought about the effect of weak disorder on the transition? Will the transition be smeared out and/or lead to intermediate phases?

Recommendation

Publish (meets expectations and criteria for this Journal)

---

## Editorial Decision

awaiting_resubmission